# HIV-1 Vpr induces cell cycle arrest and enhances viral gene expression by depleting CCDC137

**Fengwen Zhang[1], Paul D Bieniasz[1,2]***

[1]Laboratory of Retrovirology, The Rockefeller University, New York, United States; [2]Howard Hughes Medical Institute, The Rockefeller University, New York, United States

**Abstract** The HIV-1 Vpr accessory protein induces ubiquitin/proteasome-dependent degradation of many cellular proteins by recruiting them to a cullin4A-DDB1-DCAF1 complex. In so doing, Vpr enhances HIV-1 gene expression and induces (G2/M) cell cycle arrest. However, the identities of Vpr target proteins through which these biological effects are exerted are unknown. We show that a chromosome periphery protein, CCDC137/cPERP-B, is targeted for depletion by HIV-1 Vpr, in a cullin4A-DDB1-DCAF1 dependent manner. CCDC137 depletion caused G2/M cellcycle arrest, while Vpr-resistant CCDC137 mutants conferred resistance to Vpr-induced G2/M arrest. CCDC137 depletion also recapitulated the ability of Vpr to enhance HIV-1 gene expression, particularly in macrophages. Our findings indicate that Vpr promotes cell-cycle arrest and HIV-1 gene expression through depletion of CCDC137.

## Introduction

Human and simian immunodeficiency viruses (HIV-1, HIV-2 and SIVs) encode several accessory proteins; Vpr, Vpx, Vif, Nef, and Vpu. While accessory proteins are often dispensable for replication in immortalized cell lines, they are important in a physiological context and typically act by removing or displacing molecules that are deleterious to virus replication. Among the HIV-1 accessory proteins, the function of ~14 kDa Viral Protein R (Vpr) remains the most enigmatic. Replication deficits of inconsistent magnitude are evident in HIV-1 mutants lacking Vpr, particularly in primary macrophages (*Balliet et al., 1994*; *Connor et al., 1995*; *Fouchier et al., 1998*), while deletion of Vpr from SIVmac modestly attenuates pathogenesis (*Hoch et al., 1995*; *Gibbs et al., 1995*).

Vpr shares a common ancestor with an HIV-2/SIV accessory protein, Vpx. Both proteins bind to VprBP (DCAF1) and in so doing recruit the cullin 4A-containing E3 ubiquitin ligase complex (CRL4) (*Zhang et al., 2001*; *Srivastava et al., 2008*; *DeHart et al., 2007*; *Hrecka et al., 2007*; *Le Rouzic et al., 2007*; *Schröfelbauer et al., 2007*; *Tan et al., 2007*; *Wen et al., 2007*). Both Vpr and Vpx proteins are incorporated into virions through an interaction with the virion structural protein Gag (*Kondo et al., 1995*; *Kewalramani et al., 1996*). Recruitment of CRL4 by virion-associated Vpx and some SIV Vpr proteins can induce the degradation of the antiviral protein SAMHD1 shortly following viral entry (*Hrecka et al., 2011*; *Laguette et al., 2011*; *Spragg and Emerman, 2013*). However, HIV-1, HIV-2 and many SIV Vpr proteins do not exhibit SAMHD1-depleting activity. Rather, HIV-1 Vpr mediated CRL4 recruitment has different biological effects, including G2/M cell-cycle arrest of infected cells (*Connor et al., 1995*; *DeHart et al., 2007*; *Hrecka et al., 2007*; *Le Rouzic et al., 2007*; *Schröfelbauer et al., 2007*; *Tan et al., 2007*; *Wen et al., 2007*; *Jowett et al., 1995*; *Rogel et al., 1995*; *Belzile et al., 2007*) and activation of the ATR (ataxia-telangiectasia and Rad3-related)-mediated DNA damage response (DDR) (*Roshal et al., 2003*; *Zimmerman et al., 2004*; *Fregoso and Emerman, 2016*). HIV-1 gene expression is also enhanced by Vpr in some contexts,

***For correspondence:**
pbieniasz@rockefeller.edu

**Competing interests:** The authors declare that no competing interests exist.

**eLife digest** Like all viruses, the human immunodeficiency virus 1 (HIV-1) cannot replicate on its own; to multiply, it needs to exploit the molecular machinery of a cell. A set of HIV-1 proteins is vital in this hijacking process, and they are required for the virus to make more of itself. However, HIV-1 also carries accessory proteins that are not absolutely necessary for the replication process, but which boost the growth of the virus by deactivating the defences of the infected cells. Amongst these proteins, the role of Viral Protein R (Vpr for short) has been particularly enigmatic.

Previous experiments have shown that, in infected cells, Vpr is linked to several biological processes: it tags for destruction a large number of proteins, it causes the cells to stop dividing, and it encourages them to express the genetic information of the virus. How these different processes are connected and triggered by Vpr is still unknown. It particular, it remains unclear which protein is responsible for these changes when it is destroyed by Vpr.

To investigate, Zhang and Bieniasz conducted a series of experiments to spot the proteins that interact with Vpr in human cells. This screening process highlighted a protein known as CCDC137, which is depleted in cells infected by HIV-1.

To investigate the role of CCDC137, Zhang and Bieniasz decreased the levels of the protein in human cells. This stopped the cells from dividing, just like during HIV-1 infection. Destroying CCDC137 also mimicked the effects of Vpr on HIV-1 gene expression, increasing the levels of virus proteins in infected cells. Finally, Zhang and Bieniasz made a mutant version of CCDC137 that Vpr could not destroy. When infected cells carried this mutant protein, they kept on dividing as normal. Taken together, these results suggest that Vpr works by triggering the destruction of the CCDC137 protein. Overall, this work represents the first step to understand the role of CCDC137 in both infected and healthy cells.

and similarly enhanced in cells arrested in G2/M (*Connor et al., 1995*; *Goh et al., 1998*; *Yao et al., 1998*; *Gummuluru and Emerman, 1999*). Numerous cellular proteins including UNG2, the SLX4 complex, helicase-like transcription factor (HLTF), survival of motor neuron-1 (SMN1), cell division cycle associated 2 (CDCA2), and zinc finger protein 267 (ZNF267) have been reported to be depleted by Vpr (*Laguette et al., 2014*; *Hrecka et al., 2016*; *Lahouassa et al., 2016*; *Greenwood et al., 2019*), However, while depletion of some of these proteins has been reported to have modest effects on cell cycle (*Greenwood et al., 2019*), none of the previously identified Vpr target proteins have been demonstrated to be solely responsible for the G2/M-arrest and gene-expression enhancement effects of HIV-1 Vpr. Indeed, the identity of the host proteins whose Vpr-induced depletion induces cell cycle arrest, or otherwise facilitates viral propagation, remains controversial. Here, we identify a Vpr target protein, cPERP-B, also known as coiled-coil domain-containing-137 (CCDC137), whose depletion causes G2/M cell cycle arrest and enhances HIV-1 gene expression, thus recapitulating the salient biological effects of HIV-1 Vpr.

## Results

### Identification of CCDC137 as a Vpr target protein

To identify HIV-1 Vpr target proteins, we used a proximity-dependent method (*Roux et al., 2012*) in which a biotin-ligase, BirA(R118G) fused an HIV-1$_{NL4-3}$ Vpr bait was expressed in proteasome inhibitor treated cells. Biotinylated proteins were enriched using streptavidin magnetic beads, identified using mass spectroscopy and the biotinylated proteome in BirA(R118G)-Vpr, and BirA(R118G) expressing cells was compared. Multiple nuclear proteins were biotinylated in BirA(R118G)-Vpr, but not BirA(R118G) expressing, cells (*Figure 1—figure supplement 1A*, *Supplementary file 1*). Notably, Ki-67, a proliferating cell marker, was the top 'hit', with >90 Ki-67 peptides detected in replicate experiments.

However, Vpr did not induce Ki-67 depletion (*Figure 1—figure supplement 1B,C*) and shRNA-induced Ki-67 depletion did not induce G2/M arrest (*Figure 1—figure supplement 1D,E*), strongly suggesting that it is not a Vpr target protein. Based on the fact that Ki-67 was the top hit in our proximity biotinylation experiments but did not itself appear to be responsible for the Vpr-induced

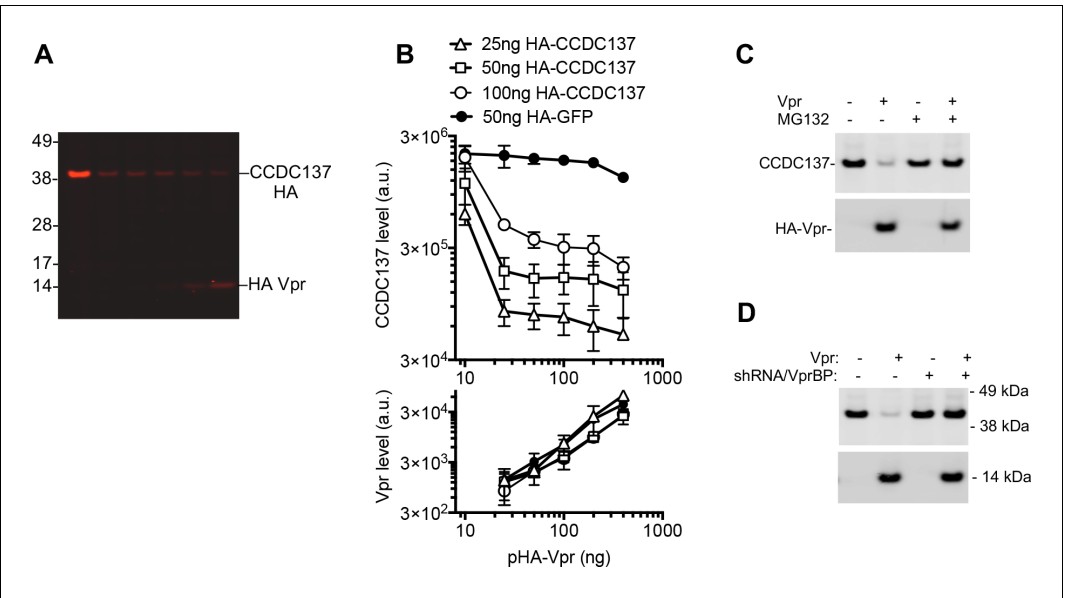

**Figure 1.** Depletion of CCDC137 by HIV-1 Vpr. (**A**) Western blot analysis of 293 T cell lysates 28 hr after transfection (100 ng ng/well) with an HA-CCDC137 expression plasmid and (0 ng, 25 ng, 50 ng, 100 ng, or 200 ng/well) of an HA tagged Vpr expression plasmid. Representative of two experiments. (**B**) Western blot quantitation of HA-CCDC137 or HA-GFP, (upper panel) and HA-Vpr (lower panel) after transfection of 293 T cells with the indicated amounts (X-axis) of HA-Vpr and target protein (CCDC137-HA [open symbols] or HA-GFP [filled symbols]) expression plasmids. The mean and range of values from two independent experiments is plotted. (**C**) Western blot analysis of 293 T cell lysates at 28 hr after transfection with 150 ng of V5-tagged CCDC137 expression plasmid and 0 ng or 30 ng of HIV-1 Vpr expression plasmid. Cells were treated with 10 μM MG132 (+) or carrier (-) for 4 hr before harvest. Representative of four experiments. (**D**) Western blot analysis of cell lysates from empty pLKO vector (-) or VprBP shRNA (+) transduced 293 T cells after transfection of 150 ng of V5-tagged CCDC137 and 30 ng of HA-Vpr expression plasmid. Representative of three experiments.
The online version of this article includes the following figure supplement(s) for figure 1:

**Figure supplement 1.** Discovery and analysis of proteins labelled with Vpr-BirA (R118G).
**Figure supplement 2.** Focused screens for Vpr target proteins.

cell cycle arrest, we hypothesized that a Ki-67-proximal or interacting protein might represent the genuine target of Vpr. Ki-67 recruits a group of proteins termed 'chromosome periphery proteins' (cPERPs), that localize within the nucleus, primarily the nucleolus, during interphase but are relocalized to chromosome peripheries during mitosis (*Booth et al., 2014*; *Ohta et al., 2010*). Therefore, we next conducted a focused screen of candidate target proteins that were either prominent hits in the BirA(R118G)-Vpr screen (*Supplementary file 1*), were nucleolar or nuclear proteins, members of the cPERP group, and/or were reported to bind Ki-67. Of numerous candidates tested in transient co-transfection/western blot assays, Vpr only induced the depletion of cPERP-B, also termed CCDC137 (*Ohta et al., 2010*; *Figure 1—figure supplement 2A–C*). To assess the potency with which Vpr could induce depletion of CCDC137, we cotransfected 293 T cells with various amounts of CCDC137 and HIV-1 Vpr expression plasmids. Both CCDC137 and Vpr proteins were tagged with the same epitope, a single copy of an HA tag, to assess their relative steady-state levels. This analysis revealed that co-expression of a barely detectable amount of Vpr resulted in the removal of much larger quantities of CCDC137, underscoring the potency with which Vpr induced CCDC137 depletion (*Figure 1A,B*, *Figure 1—figure supplement 2D*). Treatment of cells with MG132 abolished the ability of Vpr to induce CCDC137 depletion, suggesting that Vpr triggered proteasome-dependent CCDC137 degradation (*Figure 1C*). Furthermore, when DCAF1, an essential component of the CRL4 complex, was depleted by lentiviral vector-mediated RNA interference, the depletion of CCDC137 was abrogated (*Figure 1D*).

## Derivation of Vpr-resistant CCDC137 mutants

We set out to mapped the degrons through which CCDC137 is depleted by Vpr. First, CCDC137 was expressed as N-terminal (residues 1–154) and C-terminal (residues 155–289) fragments. Transfection experiments revealed that the isolated N-terminal but not the C-terminal portion was depleted by Vpr (*Figure 2—figure supplement 1A*). Alanine scanning mutagenesis in the context of full length CCDC137, whereby blocks of 5 residues were mutated throughout the N-terminal CCDC137 portion revealed that CCDC137 residues 61 to 75 were important for Vpr-induced depletion (*Figure 2A*, *Figure 2—figure supplement 1B*). Additionally, alanine substitutions of an LxxLL motif (positions 228–232) through which CCDC137 binds nuclear receptors (*Youn et al., 2018*), also reduced Vpr-induced CCDC137 depletion (L/A, mutation, *Figure 2A*). Combined substitution of

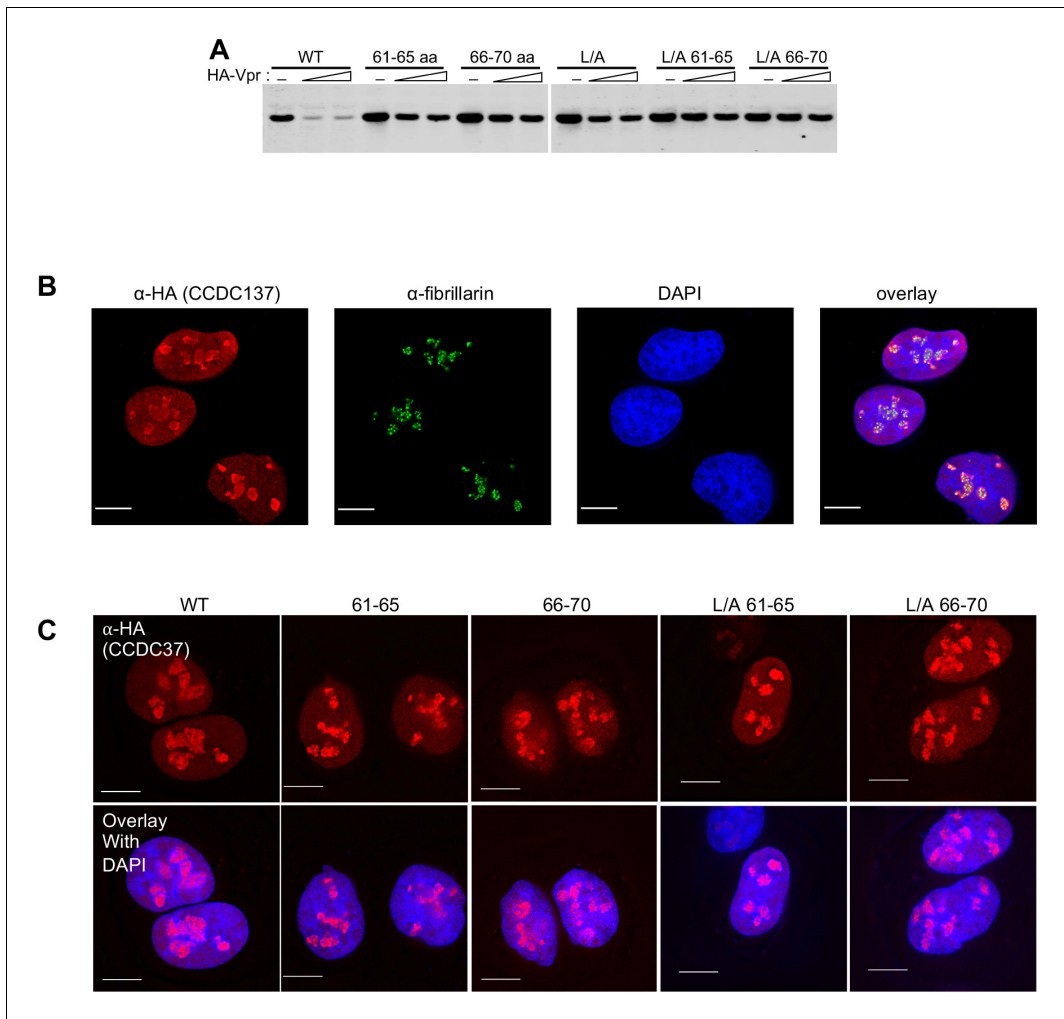

**Figure 2.** Derivation of Vpr-resistant CCDC137 mutants. (**A**) Western blot analysis of 293 T cell lysates 28 hr after transfection with plasmids expressing HA-tagged CCDC137 encoding alanine substitutions at positions 61–65 or 66–70, in the LXXLL motif (L/A), or both, along with 0 ng, 50 ng or 100 ng of an HA-Vpr expression plasmid. Representative of three experiments. (**B**) Immunofluorescent staining of U2OS cells stably expressing V5-tagged wild-type CCDC137 (red). Endogenous fibrillarin was also immunostained (green) and DNA was stained with DAPI (blue). Scale bar: 10 μm. Representative of two experiments. (**C**) Immunostaining of U2OS cells stably expressing HA-tagged wild-type (WT) or mutant CCDC137 bearing Ala substitutions at positions 61 to 65 (61-65), 66 to 70 (66-70) alone or in combination with LXXLL motif mutations (L/A). Scale bar: 10 μm. Representative of two experiments.

The online version of this article includes the following figure supplement(s) for figure 2:

**Figure supplement 1.** Mapping of degrons in CCDC137 for Vpr-induced depletion.

CCDC137 residues 61–65 or 66–70, coupled with the L/A (228-232) mutation caused CCDC137 to be substantially Vpr-resistant (*Figure 2A*). As previously reported, CCDC137 formed foci in the nucleus that colocalized with fibrillarin, a nucleolar marker, during interphase (*Ohta et al., 2010*; *Figure 2B*) and this property was unaffected by the aforementioned residue 61 to 65, 66 to 70, or the L/A (228-232) Vpr resistance-inducing substitutions (*Figure 2C*).

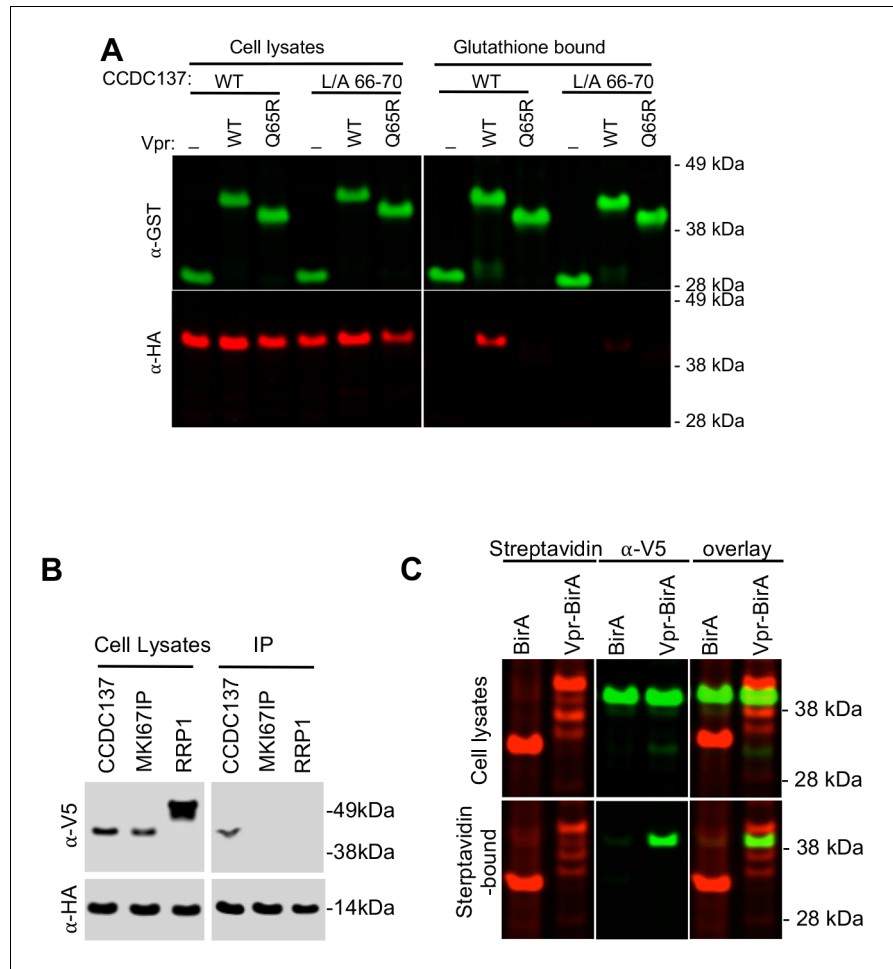

**Figure 3.** Physical association of Vpr with CCDC137. (**A**) Western analysis of cell lysates and glutathione-agarose-bound fractions following transfection of 293 T cells with plasmids expressing GST (-), GST-Vpr WT, or GST-Vpr mutant Q65R along with plasmids expressing HA-tagged wild-type (WT) or Vpr-resistant CCDC137 (L/A 66–70). Representative of three experiments. (**B**) Western blot analysis of cell lysates and immunoprecipitates (IP) following cotransfection of 293 T cells with plasmids expressing V5-tagged CCDC137, MKI67IP, or RRP1 along with a HA-Vpr expression plasmid. Immunoprecipitation was performed with an anti-HA monoclonal antibody and protein G beads. Representative of three experiments. (**C**) Western analysis of cell lysates and streptavidin bead-bound proteins following transfection of 293 T cells stably expressing V5-tagged CCDC137 with plasmids expressing BirA (R118G) or Vpr-BirA (R118G). Cells were treated with 50 μM biotin at 24 hr after transfection, and 10 μM MG132 at 40 hr post transfection, and harvested at 44 hr post transfection. Biotinylated proteins were detected with IRDye 680RD Streptavidin (red, left panel) and CCDC137 was detected with anti-V5 antibody (green, middle panel). Representative of three experiments.

The online version of this article includes the following figure supplement(s) for figure 3:

**Figure supplement 1.** Vpr (Q65R) does not deplete CCDC137 293 T cells were transfected with HA-tagged CCDC137 expression plasmid along with varying amounts (0 ng, 25 ng, or 50 ng) of HA-tagged wild-type NL4-3 Vpr (WT), mutant Q65R (Q65R), expression plasmids.

## Physical association of CCDC137 and Vpr

To determine whether Vpr and CCDC137 were physically associated, we co-expressed a glutathione S-transferase (GST)-Vpr fusion protein with CCDC137 in 293 T cells. Wild-type CCDC137, but not a Vpr-resistant CCDC137 mutant (L/A 66–70), could be co-precipitated with GST-Vpr (*Figure 3A*) when coexpressed in 293 T cells. Conversely, a mutant GST-Vpr (Q65R) that did not cause CCDC137 depletion (*Figure 3—figure supplement 1*) did not co-precipitate CCDC137 (*Figure 3A*). Similarly, V5-tagged CCDC137 could be co-immunoprecipitated with HA-tagged Vpr, but other control proteins, including Ki-67-interacting protein (MKI67IP) and nucleolar-localized ribosomal RNA processing protein 1 (RRP1), did not coimmunoprecipitate with Vpr (*Figure 3B*). When the biotinylation proximity assay (*Roux et al., 2012*) was coupled to Western blot analysis, CCDC137 was found to be biotinylated by Vpr-BirA(R118G) but not BirA(R118G) despite our inability to detect biotinylated CCDC137 in the initial mass spectroscopic screens (*Figure 3C*).

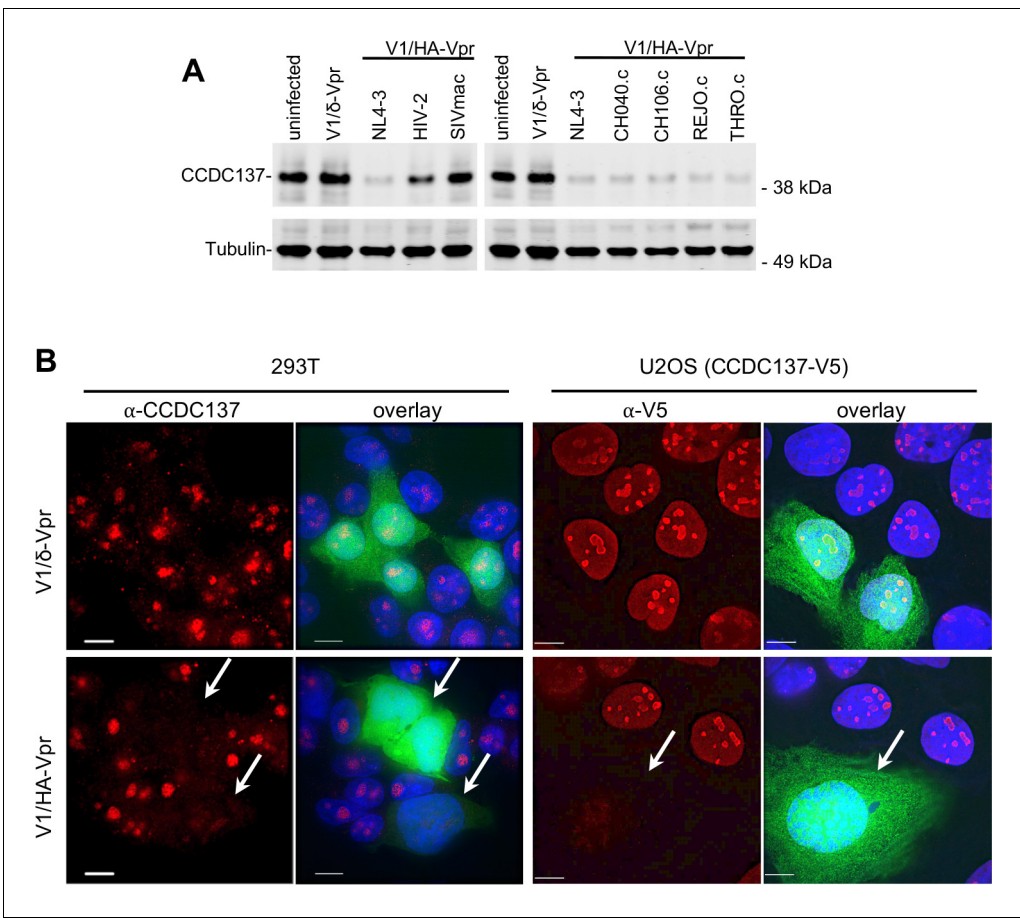

**Figure 4.** Depletion of CCD137 in HIV-1 infected cells. (**A**) Western analysis of 293 T cells 48 hr after infection (MOI = 2) with minimal HIV-1 viruses (V1) carrying no Vpr (V1/δVpr), or Vpr proteins from HIV-1$_{NL4-3}$, HIV-2, SIVmac, or HIV-1 strains (CH040.c, CH106.c, REJO.c, or THRO.c). Representative of three experiments. (**B**) Immunofluorescent detection of endogenous (293 T cells, left) or V5-tagged (U2OS cells, right) CCDC137 at 48 hr after infection with V1/δ-Vpr (upper) or V1/HA-Vpr (lower). Infected, GFP-positive are indicated by arrows. Scale bar = 10 µm. Additional examples in *Figure 4—figure supplement 2*. Representative of three experiments.
The online version of this article includes the following figure supplement(s) for figure 4:

**Figure supplement 1.** V1, a minimal HIV-1 genome.
**Figure supplement 2.** Cell cycle effects of HIV-1, HIV-2, and SIVmac Vpr.
**Figure supplement 3.** Immunofluorescent detection of CCDC137 depletion by Vpr.

## Endogenous CCDC137 depletion in HIV-1 infected cells

We infected 293T or U2OS cells with a minimal version of HIV-1 (termed V1) in which Gag, Pol, Vif, Vpu and Env carry deletions or nonsense mutations. In V1 infected cells, Tat, Rev, HA-tagged Vpr and GFP are expressed, all driven by native HIV-1 LTR sequences, and infected cells can be identified by flow cytometry or microscopy (*Figure 4—figure supplement 1*). Infection with V1 lacking Vpr (V1/δ-Vpr) had no effect on endogenous CCDC137 levels, while infection with V1/HA-Vpr, encoding Vpr from one of several HIV-1 laboratory adapted or primary transmitted founder strains, caused profound CCDC137 depletion (*Figure 4A*). HIV-2 Vpr induced partial CCDC137 depletion while SIV$_{MAC}$ Vpr did not affect CCDC137 levels (*Figure 4A*), consistent with their abilities, or lack thereof in the case of SIVmac, to induce G2M arrest (*Figure 4—figure supplement 2*). Immunofluorescent staining of endogenous CCDC137 in 293 T cells, or ectopically expressed V5-tagged CCDC137 in U2OS cells, showed that CCDC137 was diminished to nearly undetectable levels in V1/

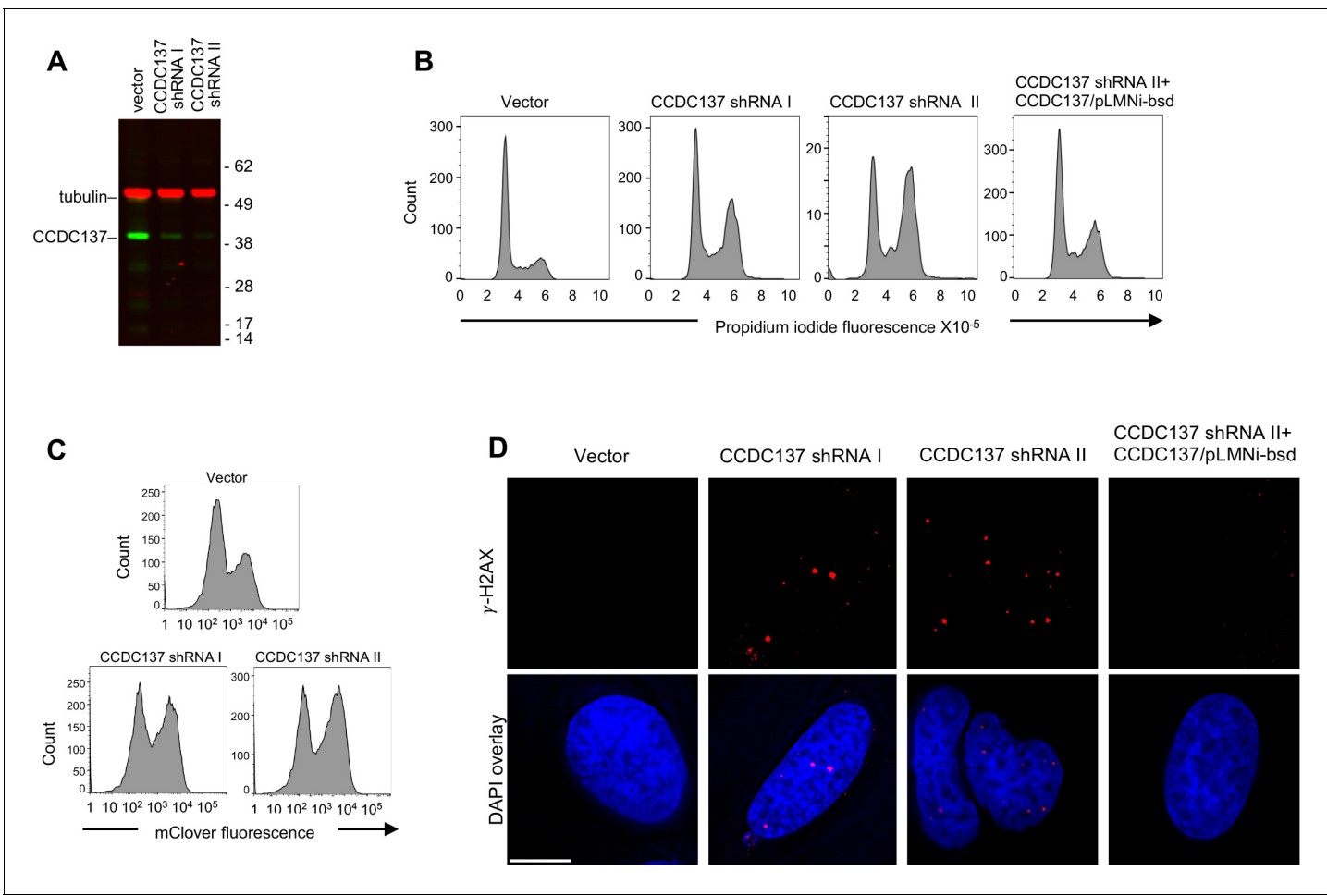

**Figure 5.** Induction of cell cycle arrest and DNA damage response by depletion of CCDC137. (**A**) Western blot analysis of 293 T cells after transduction with pLKO lentivirus vectors carrying no shRNA (vector) or an shRNAs targeting one of two sequences on CCDC137 mRNA (I, II) and selection with puromycin for 40 hr. Representative of three experiments. (**B**) Propidium iodide (PI) DNA content staining of 293 T cells transduced with shRNA-carrying lentiviruses as in (**A**). Rightmost panel, 293 T cells were also transduced with a retroviral vector (pLMNiBsd) expressing exogenous CCDC137. Representative of four experiments. (**C**) FACS analysis of a U2OS-derived cell clone stably expressing mClover-hGeminin(1–110 aa), following transduction and selection as in (**A**). Representative of two experiments. (**D**) Immunofluorescent staining of γ-H2AX foci (red) in U2OS cells following transduction as in (**B**). Scale bar: 10 μm. Representative of four experiments.

The online version of this article includes the following video and figure supplement(s) for figure 5:

**Figure supplement 1.** Cell cycle arrest induced by CCDC137 depletion.

**Figure 5—video 1.** G2/M arrest induced by CCDC137 depletion.

https://elifesciences.org/articles/55806#fig5video1

HA-Vpr infected (GFP+) cells compared to neighboring, uninfected (GFP-) cells or V1/δ-Vpr infected cells (*Figure 4B*, *Figure 4—figure supplement 3*).

## CCDC137 depletion causes G2/M cell cycle arrest and a DNA damage response

To ascertain whether CCDC137 depletion might be responsible for Vpr-induced cell cycle arrest, we next asked whether depletion of CCDC137 per se could induce G2/M cell cycle arrest. Lentiviral constructs expressing CAS9 and CCDC137-targeting CRISPR guide RNAs efficiently generated CCDC137 knockout alleles, but most transduced cells died and none of the numerous surviving cell clones that were analyzed contained frameshifting indels in both copies of CCDC137, suggesting that CCDC137 is essential for proliferating cell viability. Two CCDC137-targeting, lentiviral shRNA vectors encoding a puromycin N-acetyl-transferase caused effective short term CCDC137 depletion after puromycin selection of 293 T cells, with CCDC137/shRNAII causing more profound depletion than CCDC137/shRNAI (*Figure 5A*). Notably, CCDC137/shRNA transduced cells accumulated in G2/M (*Figure 5B*) as revealed by propidium iodine (PI) staining and the extent of CCDC137 depletion and G2/M accumulation were correlated (*Figure 5A,B*). A CCDC137 cDNA construct lacking the 3'UTR targeted by CCDC137/shRNAII substantially rescued G2/M arrest (*Figure 5B*). As an alternative way to assess the effect of CCDC137 knockdown on cell cycle, we utilized a Fluorescent Ubiquitination-based Cell Cycle Indicator (FUCCI) construct (*Sakaue-Sawano et al., 2008*), which couples the cell-cycle regulated proteins with fluorescent proteins. G2/M accumulation was also evident upon CCDC137 depletion in several U2OS cell clones expressing mClover or mKusabira-Orange2 (mKO2) fluorescent proteins fused to geminin (1–110 aa) that are depleted during G1 but present during G2/M (*Figure 5C*, *Figure 5—figure supplement 1*). Accordingly, live cell imaging of U2OS/mClover-hGeminin (1–110 aa) cells revealed fluctuating fluorescence that disappeared upon cell division, while CCDC137 depleted cells did not divide and retained mClover-hGeminin fluorescence, indicating G2/M growth arrest until apparent cell death (*Figure 5—video 1*). HIV-1 and HIV-2 Vpr are able to activate the DNA damage response (DDR) which activates response pathways through ATM/ATR and Chk1/2 kinases, leading to cell cycle arrest (*Roshal et al., 2003*; *Zimmerman et al., 2004*; *Fregoso and Emerman, 2016*). As part of the Vpr-induced DNA damage response, histone H2A variant H2AX, a marker for DNA damage, undergoes phosphorylation at Ser 139 (γ-H2AX) and forms nuclei foci. CCDC137 depletion using shRNA also caused accumulation of nuclear foci of γ-H2AX (*Figure 5D*), mimicking the reported Vpr-induced DDR. Importantly, the CCDC137 depletion-induced DDR was rescued by expression of a shRNA-resistant CCDC137 cDNA (*Figure 5D*).

## Vpr-resistant CCDC137 mutants abrogate Vpr-induced G2/M cell cycle arrest

If Vpr induces cell-cycle arrest through depletion of CCDC137, we reasoned that Vpr-induced cell-cycle arrest should be alleviated by overexpression of the Vpr-resistant CCDC137 mutants. U2OS cells containing doxyxcycline inducible CCDC137 expression constructs were treated with doxycycline to induce the expression of CCDC137 and then infected with V1/HA-Vpr or V1/δ-Vpr at an MOI of 2 for analysis of protein levels (*Figure 6A*), or an MOI of 0.5 for analysis of DNA content (*Figure 6B*). Wild-type CCDC137 was depleted following V1/HA-Vpr infection while mutant CCDC137 (L/A 66–70) largely resisted Vpr-induced depletion (*Figure 6A*). Crucially, overexpression of WT CCDC137 partly ameliorated the G2/M arrest effect of Vpr, while expression of the Vpr-resistant CCDC137 (L/A 66–70) mutant conferred nearly complete resistance to Vpr-induced G2/M arrest (*Figure 6B,C*), suggesting that depletion of CCDC137 is necessary for G2/M cell cycle arrest induction by Vpr.

## Enhancement of HIV-1 gene expression by CCDC137 depletion

Prior work has shown that Vpr enhances HIV-1 HIV-1 gene expression in a variety of cell types (*Connor et al., 1995*; *Goh et al., 1998*; *Gummuluru and Emerman, 1999*). Both of these phenotypes have also been associated with G2/M arrest properties of Vpr. We therefore assessed the effects of Vpr expression or CCDC137 depletion on HIV-1 gene expression in various cell types.

Live-cell imaging of U2OS/mClover-hGeminin cells infected with V1/δ-Vpr/mCherry resulted in accumulating mCherry fluorescence and fluctuating mClover-hGeminin fluorescence, indicative of a

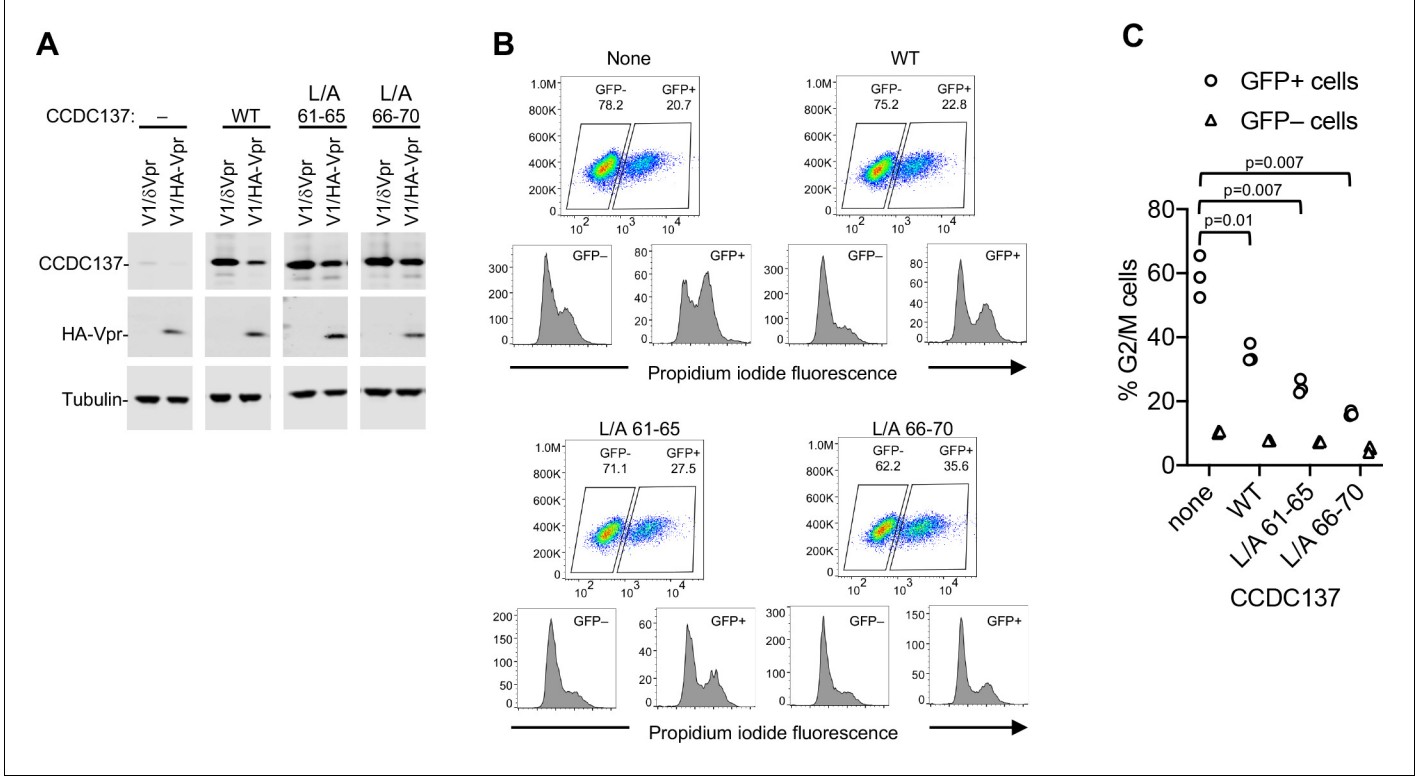

**Figure 6.** Vpr-resistant CCDC137 attenuates Vpr-induced G2/M arrest. (**A**) Western blot analysis of U2OS cells stably expressing doxycycline-inducible wild-type or Vpr depletion-resistant CCDC137 after doxycycline treatment for 24 hr and infection (MOI = 2) with V1/HA-Vpr or V1/δ-Vpr for 48 hr. Representative of three experiments. (**B**) DNA content assay of WT or mutant CCDC137-expressing cells infected (MOI = 0.3) with V1/HA-Vpr. Upper plots depict forward scatter (Y axis) vs GFP fluorescence (X-axis) and percentages of GFP+ and GFP- cells. Lower plots depict PI staining in the uninfected (GFP-, lower left) and infected (GFP+, lower right) populations. Representative of three experiments. (**C**) Quantitation and statistical analysis of the effects of WT or Vpr-resistant mutant CCDC137 overexpression on Vpr-induced cell cycle arrest. The percentage of cells in G2/M, as determined by DNA content analysis is plotted for three independent experiments. The indicted p-values were calculated for infected (GFP+) cells expressing no CCDC137 versus WT or mutant CCDC137 using a t-test with Welch's correction.

normal cell cycle. Conversely, V1/HA-Vpr/mCherry infection induced accumulation of mClover fluorescence in infected, mCherry+ cells (*Figure 7—video 1*). Notably, the level of mCherry fluorescence was clearly greater in V1/HA-Vpr/mCherry infected cells than in V1/δ-Vpr/mCherry infected cells (*Figure 7—video 1*). Moreover, live-cell imaging of U2OS/mClover-hGeminin cells transduced with a CCDC137-depleting shRNA vector (CCDC137/shRNAII, *Figure 5A*) and then infected with V1/δ-Vpr/mCherry, showed that CCDC137 depletion using shRNA recapitulated the effect of Vpr, causing both G2M arrest and increased HIV-1 gene (mCherry) expression (*Figure 7—video 2*). This finding was reproduced using FACS analysis of V1/δ-Vpr/GFP-infected U2OS cells. Specifically, prior transduction of U2OS cells with a CCDC137-depleting shRNA vector (CCDC137/shRNAII, *Figure 5A*) caused elevated GFP expression in infected cells (*Figure 7A,B*).

## CCDC137 depletion increases HIV-1 gene expression in primary cells

While Vpr expression caused CCDC137 depletion and increased HIV-1 gene expression in U2OS cells, these cells are not natural targets of HIV-1. Therefore, we next examined the effects of Vpr on CCDC137 and HIV-1 gene expression in primary cells.

Immunofluorescence assays in V1/HA-Vpr and V1/δ-Vpr infected primary macrophages showed that Vpr effectively depleted CCDC137 therein while the Q65R mutant Vpr did not (*Figure 8A*, *Figure 8—figure supplement 1*). Infection of primary CD4+ T-cells with V1/HA-Vpr and V1/δ-Vpr revealed that the presence of Vpr resulted in ~2 fold higher levels of GFP expression than did in its absence (*Figure 8B,C*). The effect of Vpr on HIV-1 gene expression was more pronounced in primary macrophages, where V1/HA-Vpr but not V1/HA-Vpr(Q65R) resulted in higher levels of GFP

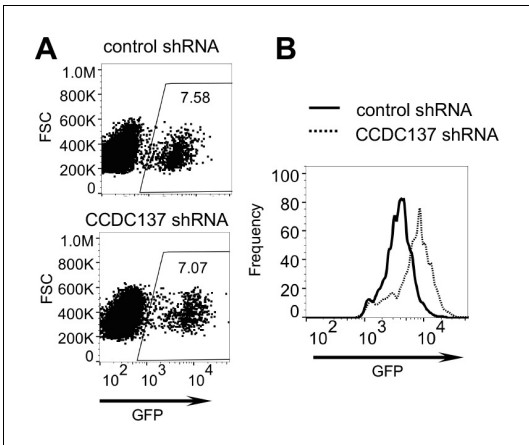

**Figure 7.** CCDC137 depletion increases HIV-1 gene expression in U2O2 cells. (**A, B**) HIV-1(GFP) expression in U2OS cells transduced with an empty lentiviral vector (solid line) or CCDC137-targeting shRNA (dotted line) and selected with puromycin prior to infection with V1/δ-Vpr/GFP for two days. (**B**) represents histogram of GFP fluorescence in infected cells, gated as shown in (**A**). Numbers in (**A**) represent % of cells within the gate. Representative of three experiments.

The online version of this article includes the following video(s) for figure 7:

**Figure 7—video 1.** G2/M arrest and HIV-1 gene expression induced by Vpr in U2OS cells.

https://elifesciences.org/articles/55806#fig7video1

**Figure 7—video 2.** HIV-1 (V1/mCherry) expression in CCDC137 depleted U2OS cells.

https://elifesciences.org/articles/55806#fig7video2

expression than did infection with V1/δ-Vpr, even though marked donor-to-donor variation was evident (*Figure 8D,E*, *Figure 8—figure supplement 1*, *Figure 8—videos 1*, *2*). Several different Vpr proteins from primary HIV-1 strains caused increased HIV-1 gene expression in macrophages (*Figure 8F*) and similar Vpr-induced enhancement of HIV-1 gene expression was evident in macrophages infected with a full-length reporter virus (HIV-1$_{NHG}$, *Figure 8—figure supplement 2*), as previously reported (*Connor et al., 1995*). Notably, the Vpr-induced increase in GFP levels in V1 infected macrophages was accompanied by elevated HIV-1 RNA levels, as assessed by in-situ hybridization (*Figure 8G,H*) indicating that Vpr enhances viral gene expression primarily by enhancing RNA synthesis or increasing RNA stability.

To enable Vpr-independent CCDC137 depletion in infected primary cells, while simultaneously measuring HIV-1 gene expression, we constructed a derivative of V1/δ-Vpr carrying an shRNA expression cassette (V1/sh, *Figure 9—figure supplement 1*). Infection with V1/sh enabled effective Vpr-independent CCDC137 depletion in infected macrophages (*Figure 9A*, *Figure 9—figure supplement 2*). Importantly, Vpr-independent CCDC137 depletion in V1/sh infected primary CD4+ T-cells resulted in an enhancement of HIV-1 gene expression (*Figure 9B*, *Figure 9—figure supplement 3A*). More strikingly, Vpr-independent CCDC137 depletion in V1/sh infected macrophages caused pronounced enhancement of HIV-1 reporter gene expression as assessed by live cell imaging or FACS assays (*Figure 9C*, *Figure 9—figure supplements 2* and *3B,C*, *Figure 9—videos 1*, *2*). This enhancing effect of CCDC137 depletion on HIV-1 gene expression was also evident when HIV-1 RNA levels were assessed by in-situ hybridization assays using probes targeting GFP (*Figure 9D,E*). Similarly, qRT-PCR (*Figure 9F*) assays of GFP and Gag mRNA levels in multiple donors, indicated that CCDC137 depletion recapitulated the effect of Vpr on HIV-1 transcription or RNA stability. Overall, in three different cell types, shRNA-driven CCDC137 depletion had similar enhancing effects on HIV-1 gene expression as did Vpr expression.

## Discussion

A persistent paradox in the study of Vpr is that Vpr-defective viruses are often selected in long term replication experiments or in chronically infected cells (*Jowett et al., 1995*; *Rogel et al., 1995*). Presumably therefore, Vpr is deleterious to HIV-1 replication in these contexts, despite its positive effects on HIV-1 gene expression. This effect seems likely to be explained the interplay of infected cell life span and burst size (*Goh et al., 1998*). In cell culture, the longevity of an infected cell plays a crucial role in determining burst size, and thus the effect of Vpr in curtailing the lifespan of an infected cell is expected to confer a competitive disadvantage. Conversely, the life span of infected T-cells in vivo is likely limited by factors other than Vpr and even a modest Vpr-induced enhancement in viral gene expression should confer advantage. Macrophages are not cycling cells, and lack ATR, Rad17 and Chk1 (*Zimmerman et al., 2006*). Thus, G2/M arrest and DDR induction by Vpr are likely not relevant therein. We confirmed herein that the Vpr-induced enhancement of HIV-1 gene expression is particularly evident in macrophages (*Connor et al., 1995*). We surmise that the key

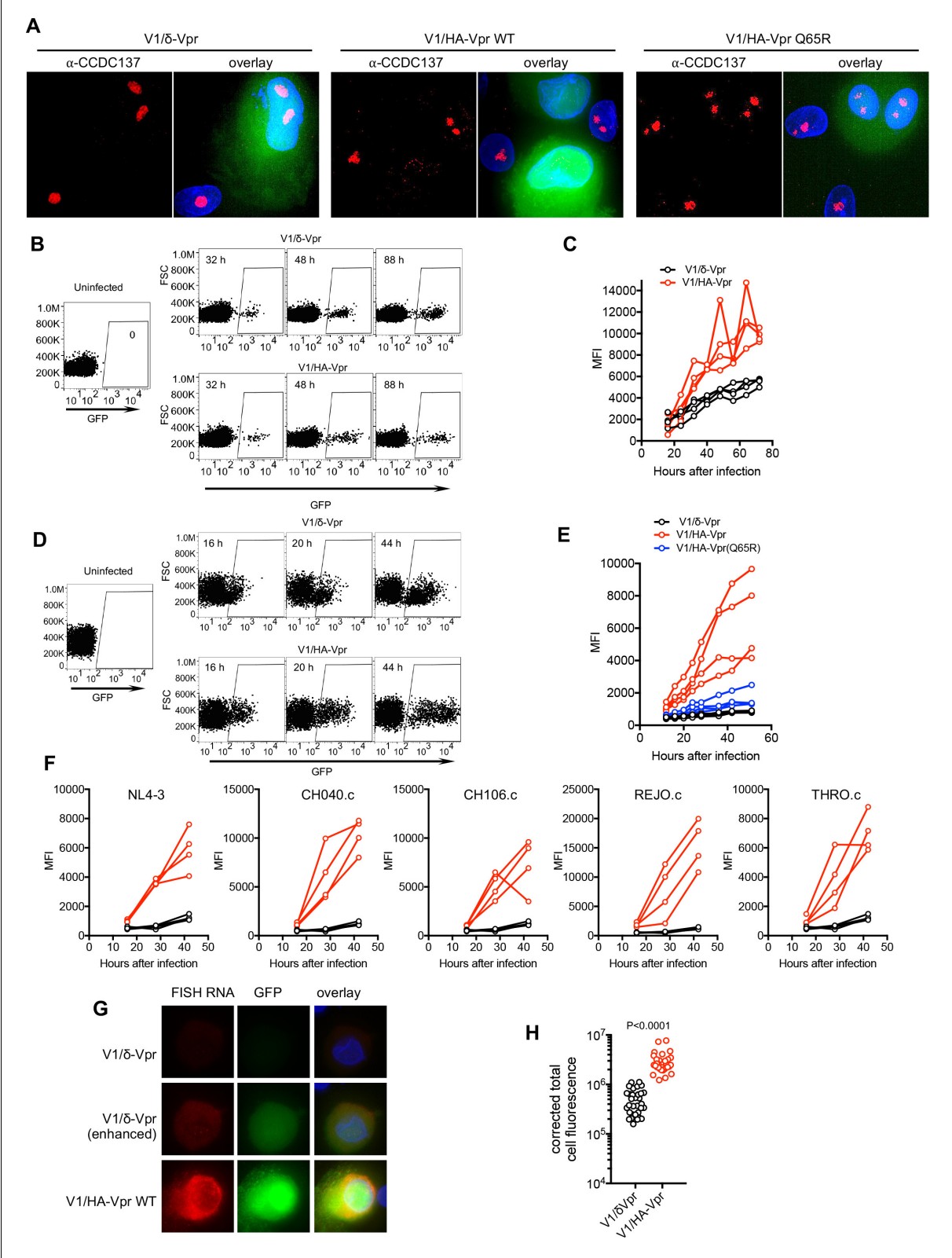

**Figure 8.** Vpr increases HIV-1 gene expression in primary cells. (**A**) Example of immunofluorescent staining to detect endogenous CCDC137 and GFP expression in primary macrophages at 48 hr after infection with V1/δ-Vpr (left), V1/HA-Vpr (center) or V1/HA-Vpr (Q65R) (right) at low MOI. Scale bar: 10 μm. Additional examples are shown in *Figure 8—figure supplement 1*. Representative of two experiments, each with three macrophage donors. (**B**) FACS analysis of GFP expression in activated primary CD4+ cells after infection with V1/δ-Vpr or V1/HA-Vpr. A representative donor is shown.
*Figure 8 continued on next page*

*Figure 8 continued*

Representative of three experiments, each with three or four donors. (C) GFP levels (mean fluorescence intensity (MFI), gated on infected cells) in primary CD4+ cells from four donors after infection with V1/δ-Vpr or V1/HA-Vpr. Representative of three experiments, each with three or four donors. (D) FACS analysis of GFP expression in macrophages after infection with V1/δ-Vpr or V1/HA-Vpr. A representative donor is shown. Representative of five experiments, each with three or four donors. (E) GFP levels (MFI, gated on infected cells) in macrophages from four donors after infection with V1/δ-Vpr, V1/HA-Vpr or V1/HA-Vpr(Q65R). Representative of five experiments, each with three or four donors. (F) FACS analysis of GFP expression macrophages after infection with V1/δ-Vpr (Black) or V1/HA-Vpr (Red) derivatives encoding Vpr proteins from several different transmitted founder virus strains. The mean fluorescent intensity (MFI) of infected cells, gated as in (D) for four macrophage donors is plotted. Representative of two experiments, each with three or four donors. (G) Representative images of primary macrophages infected with V1/δ-Vpr or V1/HA-Vpr and subjected to fluorescent in situ hybridization (FISH) based detection of HIV-1 RNA using probes directed at the GFP sequence. The FISH signal is displayed in red and GFP protein signal is displayed in green. The upper (V1/δ-Vpr) and lower (V1/HA-Vpr) rows are displayed with the same gain and brightness/contrast settings. The center (V1/δ-Vpr) row is a duplicate of the upper row, displayed with enhanced brightness to enable visualization of the FISH and GFP signals. Representative of two experiments, each with three donors. (H) GFP RNA levels (total cell fluorescence) determined by FISH analysis of macrophages infected with V1/δ-Vpr or V1/HA-Vpr. Each symbol represents a single cell from a representative donor, from one of two experiments, each with three donors. P-value is calculated using a Welch's t-test.

The online version of this article includes the following video and figure supplement(s) for figure 8:

**Figure supplement 1.** Immunofluorescent detection of CCDC137 depletion by Vpr and increased V1/GFP expression in macrophages.

**Figure supplement 2.** Enhancement of HIV-1 gene expression in macrophages by Vpr.

**Figure 8—video 1.** HIV-1 (V1/δ-Vpr vs V1/HA-Vpr) expression in macrophages (donor #1).

https://elifesciences.org/articles/55806#fig8video1

**Figure 8—video 2.** HIV-1 (V1/δ-Vpr vs V1/HA-Vpr) expression in macrophages (donor #2).

https://elifesciences.org/articles/55806#fig8video2

consequence of Vpr-induced CCDC137 depletion is enhancement of HIV-1 gene expression, with G2/M arrest and DDR in cycling cells perhaps representing an epiphenomenon.

While this work was in progress, Greenwood et al reported the effects of HIV-1 infection on the cellular proteome and found that many proteins were depleted from HIV-1 and HIV-2 infected cells, in a Vpr-dependent manner (*Greenwood et al., 2019*). Consistent with our findings, CCDC137 was among the proteins found to be depleted in HIV-1 and HIV-2 infected cells. While CCDC137 was not investigated further, Greenwood et al reported that depletion of other apparent Vpr targets, (SMN1, CDCA2 and ZNF267) caused a degree of G2/M arrest. While we cannot exclude the possibility that additional target proteins contribute to the G2/M arrest properties of Vpr, the data presented herein suggests that CCDC137 represents the dominant mediator of the G2/M arrest and HIV-1 gene expression effects of HIV-1 Vpr. Further work, including a detailed analysis of Vpr mutants with distinct target protein specificities should help delineate the relative contributions of various target proteins to Vpr-associated phenotypes. Of note, other viral genes, in particular Vif, are able to affect the cellular environment, including cell cycle perturbation. To exclude the influence of other viral genes, and to facilitate shRNA experiments in primary cells, we employed a minimal version of HIV-1 in most experiments to study the effects of Vpr on cell cycle and gene expression in the target cells. However, we note that the effects of Vpr on gene expression in macrophages were recapitulated with a near full length viral construct, and are consistent with data obtained using a different near full length viral reporter construct reported by *Connor et al., 1995*.

That Vpr associates with a cPERP protein is consistent with prior findings that Vpr binds to chromatin and, with VprBP, forms chromatin-associated nuclear foci, a property that is associated with G2/M cell cycle arrest (*Lai et al., 2005*; *Belzile et al., 2010*). CCDC137 is a poorly characterized protein - little is known about its function. CCDC137 has been reported to sequester retinoic acid receptor (RAR) to the nucleolus and thus is dubbed RaRF (Retinoic acid Resistance Factor) (*Youn et al., 2018*) but whether this property is relevant to HIV-1 gene expression is unknown. Further work will be required to discern the mechanistic details of how CCDC137 affects G2/M transition and inhibits HIV-1 gene expression. Nevertheless, the findings reported herein reveal an important aspect of how HIV-1 manipulates host cells to facilitate its replication.

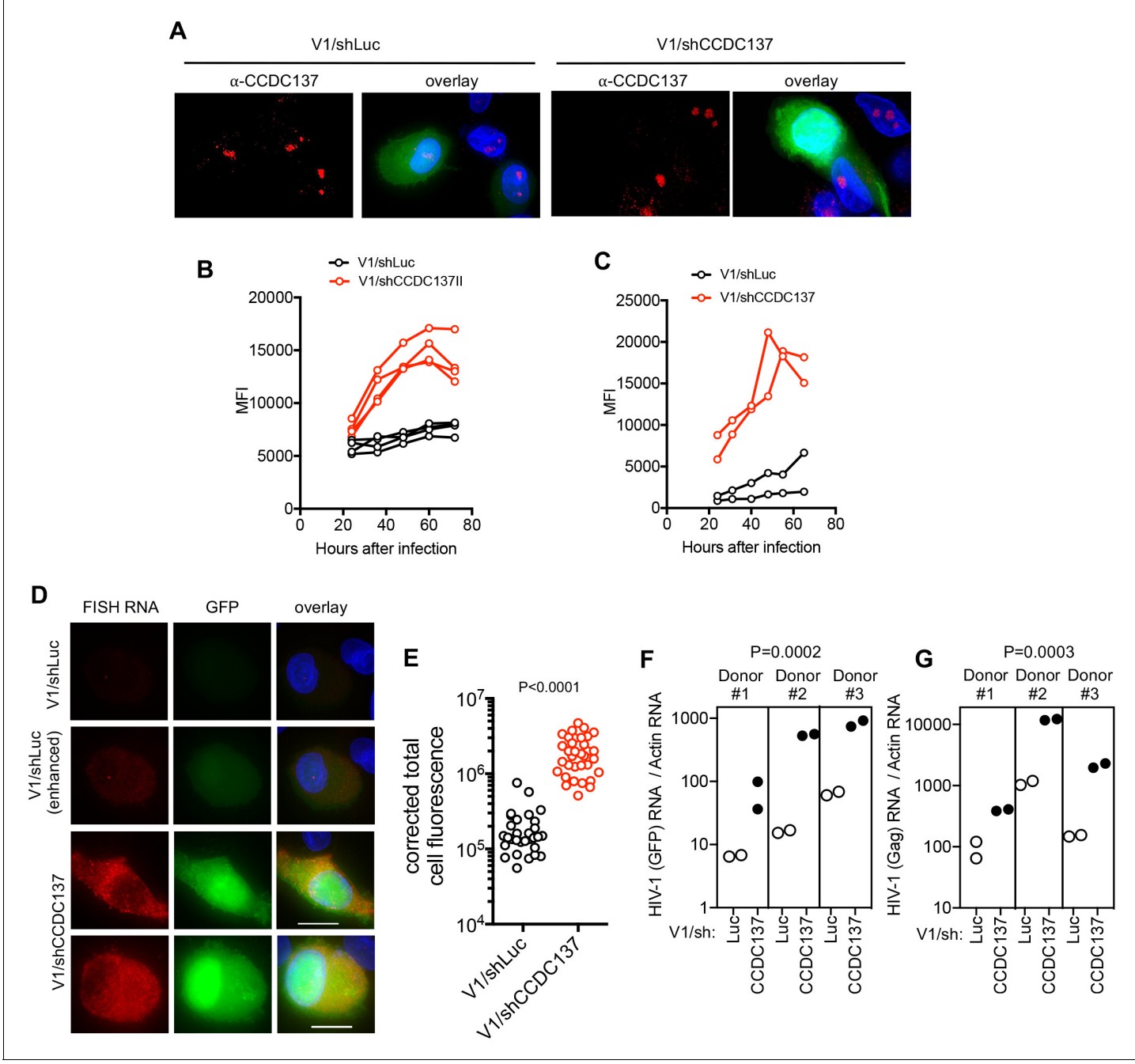

**Figure 9.** CCDC137 depletion increases HIV-1 gene expression in primary cells. (**A**) GFP expression and Immunofluorescent staining to detect endogenous CCDC137 in primary macrophages at 48 hr after infection with V1/sh (left) or V1/shCCDC137 II (right) at low MOI. Scale bar: 10 μm. Additional examples are shown in *Figure 9—figure supplement 2*. Representative of two experiments, each with three donors. (**B**) GFP expression in primary CD4+ T-cells after infection with V1/shLuc or V1/shCCDC137. MFI (gated on infected cells) for four donors is shown. Representative of three experiments, each with three or four donors. (**C**) GFP expression in macrophages after infection with V1/shLuc or V1/shCCDC137. MFI of infected cells for two donors is shown. Representative of three experiments, each with two to four donors. (**D**) Representative images of primary macrophages infected with V1/shLuc or V1/shCCDC137II and subjected to fluorescent in situ hybridization (FISH) based detection of HIV-1 RNA using probes directed at the GFP sequence. The FISH signal is displayed in red and GFP protein signal is displayed in green. The upper (V1/shLuc) and lowest two (V1/shCCDC137II) rows are displayed with the same gain and brightness/contrast settings. The second (V1/shLuc) row is a duplicate of the upper row, displayed with enhanced brightness to enable visualization of the FISH and GFP signals. Representative of two experiments, each with three donors. (**E**) GFP RNA levels (total cell fluorescence) determined by FISH analysis of macrophages infected with V1/shLuc or V1/shCCDC137. Each symbol represents a single cell from a representative donor from one of two experiments, each with three donors. P-value is calculated using a Welch's t-test. (**F**) qRT-PCR measurement of HIV-1 RNA (GFP) levels in three macrophage donors after infection with V1/shLuc or V1/shCCDC137II. Each symbol

*Figure 9 continued*

represents a technical replicate. Representative of two experiments, each with three donors. P-value is calculated using a ratio paired t-test for the three displayed donors. (**G**) qRT-PCR measurement of HIV-1 RNA (Gag) levels in three macrophage donors after infection with V1/shLuc or V1/shCCDC137II. Each symbol represents a technical replicate. Representative of two experiments, each with three donors. P-value is calculated using a ratio paired t-test for the three displayed donors.

The online version of this article includes the following video and figure supplement(s) for figure 9:

**Figure supplement 1.** V1/sh, a minimal HIV-1 genome carrying a reporter gene and an shRNA expression cassette.
**Figure supplement 2.** Immunofluorescent detection of CCDC137 depletion by shRNA and increased.
**Figure supplement 3.** Enhancement of HIV-1 gene expression in macrophages and CD4+ T-cells by shRNA-mediated CCDC137 depletion.
**Figure 9—video 1.** HIV-1 (V1/shLuc vs V1/shCCDC137) expression in macrophages (donor #3).
https://elifesciences.org/articles/55806#fig9video1
**Figure 9—video 2.** HIV-1 (V1/shLuc vs V1/shCCDC137) expression in macrophages (donor #4).
https://elifesciences.org/articles/55806#fig9video2

# Materials and methods

## Key resources table

| Reagent type (species) or resource | Designation | Source or reference | Identifiers | Additional information |
|---|---|---|---|---|
| Gene (*Homo sapiens*) | CCDC137 | This paper | NCBI Gene ID: 339230 | |
| Cell line (*Homo-sapiens*) | 293T | ATCC | CRL-3216 | |
| Cell line (*Homo-sapiens*) | U2OS | ATCC | HTB-96 | |
| Antibody | anti-FLAG | Sigma | Cat# F3165 | WB (1:1000) Monoclonal ANTI-FLAG M2 antibody produced in mouse |
| Antibody | anti-HA | BioLegend | Cat# 901515 | WB (1:1000) Anti-HA.11 Epitope Tag Antibody |
| Antibody | anti-Ki67 | Abcam | Cat# ab16667 | IF (1:500); WB (1:1000) Rabbit Anti-Ki67 antibody [SP6] |
| Antibody | anti-V5 | Invitrogen | Cat# R960-25 | WB (1:1000); IF (1:200) Mouse Monoclonal |
| Antibody | anti-V5 | Thermo Fisher Scientific | Cat# PA1-993 | WB (1:1000) Rabbit Polyclonal |
| Antibody | anti-CC137 | Abcam | Cat# ab185368 | WB (1:500); IF (1:200) Rabbit polyclonal to CC137 |
| Antibody | anti-CC137 | Abcam | Cat# ab183864 | WB (1:500) Rabbit polyclonal to CC137 |
| Antibody | anti-gamma H2AX phosphor S139 | Abcam | Cat# ab11174 | IF (1:500) Rabbit polyclonal to gamma H2A.X (phospho S139) |
| Recombinant DNA reagent | V1/δ-Vpr | This paper | | a minimal proviral plasmid |
| Recombinant DNA reagent | V1/HA-Vpr | This paper | | a minimal proviral plasmid expressing HA-tagged Vpr |
| Recombinant DNA reagent | V1/Vpr | This paper | | a minimal proviral plasmid expressing Vpr |

*Continued on next page*

*Continued*

| Reagent type (species) or resource | Designation | Source or reference | Identifiers | Additional information |
|---|---|---|---|---|
| Recombinant DNA reagent | V1/mCherry | This paper | | mCherry replaces GFP in V1 |
| Recombinant DNA reagent | pLKO.1 TRC | Addgene | Cat# 10878 | Lentiviral construct to express the shRNA. |
| Recombinant DNA reagent | pLKOΔ-puro | *Busnadiego et al., 2014* | | LKO-derived lentiviral expression vector |
| Recombinant DNA reagent | pLKOΔ-puro CCDC137 | This paper | | tetracycline-inducible CCDC137 expression pLKO vector |
| Peptide, recombinant protein | GM-CSF | Thermo Fisher | Cat# PHC2011 | Macrophage differentiation |
| Commercial assay or kit | FxCycle PI/RNase Staining Solution | Invitrogen | Cat#: F10797 | |
| Commercial assay or kit | Power SYBR Green RNA-to-CT 1-Step Kit | Thermo Fisher | Cat# 4389986 | |
| Chemical compound, drug | MG 132 | Sigma Aldrich | Cat#:M7449 | |
| Software, algorithm | MetaMorph | Molecular Devices | | |
| Software, algorithm | Prism | Graphpad | | |

## Antibodies

Monoclonal antibodies used herein included anti-FLAG (Sigma), anti-HA (BioLegend anti-HA.11), anti-HIV capsid p24 (183-H12-5C, NIH AIDS Research and Reference Reagent Program), anti-Ki67 (abcam, ab16667), anti-tubulin (Sigma, T9026), and anti-V5 (Invitrogen, R960-25). Polyclonal antibodies were purchased from ThermoFisher (anti-V5, PA1-993), or Abcam (anti-CC137, ab185368 or ab183864; anti-gamma H2AX phosphor S139, ab11174; anti-Fibrillarin, ab5821). Secondary antibodies included goat anti-mouse or anti rabbit IgG conjugated to Alexa Fluor 488 or Alexa Fluor 568 (Invitrogen) for immunostaining, or IRDye 800CW and IRDye 680, or IRDye 680RD Streptavidin (LI-COR Biosciences) for Western blot analysis.

## Plasmid construction

An HA-epitope was fused in-frame at the N-terminus of HIV-1$_{NL4-3}$ Vpr and subcloned into pCR3.1 for transient cotransfection experiments. To express Vpr driven by an HIV-1 long terminal repeat (LTR) in the context of a viral genome, a proviral plasmid was generated harboring a minimal viral genome (V1) (*Zennou and Bieniasz, 2006*), engineered from HIV-1$_{NL4/3}$ (R7/3) harboring large deletions or inactivating mutations in (Gag, Pol, Vif, Vpu and Env) and in which Nef is replaced with GFP. Sequences encoding an HA epitope were fused in frame at N-terminus of Vpr to generate V1/HA-Vpr while V1/δ-Vpr was constructed by deletion of the nucleotide sequences between the cPPT/FLAP and the SalI site within Vpr. In some constructs, the WT HIV-1$_{NL4-3}$ Vpr was replaced with Vpr-encoding sequences from Q65R NL4-3 Vpr, transmitted HIV-1 founder strains, HIV-2, SIV$_{AGM}$ Sab, or SIVmac. To construct the V1/HA-Vpr or V1/δ-Vpr expressing mCherry (V1/mCherry), an open reading frame encoding mCherry was amplified, digested with NotI/XhoI, and inserted into V1 to replace GFP. To construct the V1/shCCDC137II carrying an shRNA targeting CCDC137, the DNA sequence containing the U6 promoter and shRNA targeting CCDC137 (I: CAGATGCTGCGGATGCTTCT; II: GGTGAAACATGATGACAACA) was PCR amplified and, after digestion with KpnI, subcloned into V1 carrying inactivating mutations at the 5' end of Vpr (ATGGAACAA/GTGGAATAA), upstream 5' of the RRE (Fig. S6). The control V1/shLuc vector contained the DNA sequence carrying U6 promoter-shRNA targeting luciferase (CGCTGAGTACTTCGAAATGTC) at the same position. V1 derivatives expressing BirA (R118G) and BirA (R118G)-Vpr were constructed by insertion of nucleotide sequences encoding BirA (R118G) or BirA (R118G)-Vpr fusion proteins into the GFP (Nef) position in V1 vector.

Plasmids expressing the various human proteins (CREB1, CREB3L1, hnRNP D, hnRNP F, POC1A, PPM1G, CCDC137, PES1, WDR18, WDR74, MAK16, NOC2L, RRS1, NIP7, hnRNPA1, hnRNP H, hnRNP K, hnRNP R, hnRNP U, hnRNP C, hnRNP A2B1, and DHX9) with a V5 epitope fused to the C-terminus were from a pCSGW-based human ORF lentiviral library. Plasmid pCR3.1 was used to express Bop1, NCL, MKI67IP, or WDR12 with an Flag epitope fused to their N-termini or to express hnRNP A2B1 or DHX9 with an HA epitope fused to their N-termini. Subsequently, CCDC137 was subcloned into pLNCX2 expression vector with an HA epitope fused to its C-terminus. CCDC137 alanine scanning mutants were generated by overlap-extension PCR amplification, using the wild-type CCDC137-HA expression plasmid as the template and inserted into the pLNCX2 expression vector. All aforementioned plasmids were constructed using PCR, Accession numbers for cDNA sequences and oligonucleotides listed in *supplementary file 2*.

GST-Vpr expression plasmids were based on pCAGGS and were constructed by PCR amplification of the Vpr coding region from HIV-1$_{NL4-3}$ which was then inserted in-frame at the 3' end of GST encoding sequences.

To construct a tetracycline-inducible CCDC137 expression vector, wild-type and mutant CCDC137 were amplified, using the corresponding pCR3.1 expression plasmids as templates, and inserted into LKO-derived lentiviral expression vector pLKOΔ-puro (*Busnadiego et al., 2014*) which also included a puromycin resistance cassette. All cloned coding sequences were verified by DNA sequencing, oligonucleotide sequences used in construction are listed in *supplementary file 2*.

For knockdown experiments, the lentiviral vector pLKO.1-TRC (*Moffat et al., 2006*) was used to deliver shRNAs. For CCDC137 shRNAs, the lentiviral vector contains two functional elements, shRNA targeting sequences (I: CAGATGCTGCGGATGCTTCT; II: GGTGAAACATGATGACAACA) and puromycin-resistance cassette.

## Cell lines

Human embryonic kidney HEK-293T (ATCC CRL-3216) were maintained in DMEM supplemented with 10% fetal bovine serum (Sigma F8067) and gentamycin (Gibco). Human bone osteosarcoma epithelial cells (U2OS, ATCC HTB-96) were grown in McCoy's 5a Medium Modified (ATCC 30–2007)/ 10% FCS/gentamycin. MT4 cells (RRID:CVCL_2632) were maintained in RPMI supplemented with 10% fetal bovine serum (FCS) and gentamycin. All cell lines used in this study were monitored by SYBR Green real-time PCR RT assay periodically to ensure the absence of retroviral contamination and stained with DAPI to ensure absence of mycoplasma. Cell line identification was documented by the suppliers. To construct cell-cycle reporter cell line, U2OS cells were transduced with a retroviral vector (pLHCX) encoding Clover (a rapidly-maturing green/yellow fluorescent protein) or mKusabira-Orange2 (mKO2, an orange fluorescent protein) fused to the N-terminus of Geminin 1–110 aa. Single cell clones were isolated after hygromycin selection.

## Primary cells

Human lymphocytes were prepared from Leukopaks from NY Blood Center by spinning on top of lymphocyte separation medium (Corning). Macrophages were then isolated by plastic adherence and differentiated using GM-CSF (Thermo Fisher). CD4+ T cells were isolated using an EasySep kit (StemCell) and maintained in RPMI/10%FCS supplemented with IL2.

## Transfection experiments

For transfection experiments in 293 T cells, cells were seeded at a concentration of $1.5 \times 10^5$ cells/ well (24-well plate), $3 \times 10^5$ cells/well (12-well plate) or $2 \times 10^6$ (10 cm dish) and transfected on the following day using polyethylenimine (PolySciences).

To test whether Vpr could induce depletion of proteins, 293 T cells in 24-well plates were transfected with 200 ng of pCR3.1-based plasmids expressing Flag-tagged proteins or HA-tagged proteins, or pCSGW-based plasmids expressing V5-tagged proteins (from human ORFs lentiviral library), along with increasing amounts (0 ng, 25 ng, or 50 ng) of a pCR3.1/HA-Vpr expression plasmid. The total amount of DNA was held constant by supplementing the transfection with empty expression vector. Cells were harvested at 28 hr post transfection and subjected to Western blot analysis.

To assess the potency with which HIV-1 Vpr induced CCDC137 depletion, 293 T cells in 24-well plates were transfected with varying amounts (0 ng, 100 ng, 200 ng, or 400 ng/well) of a pCR3.1/CCDC137-HA expression plasmid and increasing amounts (0 ng, 25 ng, 50 ng, 100 ng, or 200 ng/well) of a pCR3.1/HA-Vpr expression plasmid. The total amount of DNA was held constant by supplementing the transfection with empty expression vector.

## Generation of HIV-1 and lentiviral vector stocks

To generate V1-derived viral stocks, 293 T cells were transfected with 5 µg of pV1-derived proviral plasmids encoding no Vpr or the various HA-tagged Vpr proteins, 5 µg of an HIV-1 Gag-Pol expression plasmid (pCRV1/GagPol) and 1 µg of VSV-G expression plasmid into 293 T cells in 10 cm dishes. Virus-containing supernatant was collected and filtered (0.2 µm) 2 days later. The lentiviral vectors that transduced CCDC137 cDNAs or shRNAs were similarly generated, except that pLKOΔ-puro or pLKO.1-TRC-derived plasmids were used in place of pV1-derived plasmids.

## Identification of Vpr proximal and interacting proteins

Ten million MT4 cells were transduced with V1-based constructs expressing BirA (R118G) or BirA (R118G)-Vpr at an MOI of 2, treated with 50 µM biotin (Sigma) and, after treatment with 10 µM MG132 for 4 hr, cells were harvested 48 hr after transduction. Biotinylated proteins were purified using a previously described protocol (*Roux et al., 2012*). In brief, cells were lysed in lysis buffer (50 mM Tris, pH 7.4, 500 mM NaCl, 0.4% SDS, 5 mM EDTA, 1 mM DTT, and 1x complete protease inhibitor [Roche]) and sonicated. After addition of Triton X-100 and further sonication, cell lysates were centrifuged and cleared supernatants were incubated with Dynabeads (MyOne Steptavadin C1 [Invitrogen]) for 4 hr. Beads were then washed with 2% SDS, followed by wash thoroughly with buffer 2 (0.1% deoxycholate, 1% Triton X-100, 500 mM NaCl, 1 mM EDTA, and 50 mM Hepes, pH 7.5), buffer 3 (250 mM LiCl, 0.5% NP-40, 0.5% deoxycholate, 1 mM EDTA, and 10 mM Tris, pH 8.1) and buffer 4 (50 mM Tris, pH 7.4, and 50 mM NaCl). Biotinylated proteins were eluted from the beads with NuPAGE LDS sample buffer supplemented with 200 µM biotin and separated by NuPAGE Bis-Tris Gels. Each gel lane was cut in five bands and subjected to LC-MS/MS analysis (Proteomics Resource Center, Rockefeller University).

## Immunoprecipitation

HEK-293T cells were transiently transfected with plasmids expressing HA-Vpr and V5-tagged human protein factors, and treated with 10 µM MG132 for 4 hr before harvest and lysis with ice-cold lysis buffer (50 mM Tris, pH 7.4, 150 mM NaCl, 0.5 mM EDTA, 1% digitonin [Sigma], supplemented with 1X complete protease inhibitor [Roche]). After lysis on ice for 10 min, followed by centrifugation at 10,000 rpm for 10 min at 4°C, clarified lysates were mixed with 1 µg anti-HA monoclonal antibody and rotated with 30 µl pre-equilibrated Protein G Sepharose 4 Fast Flow resin (GE healthcare) for 3 hr at 4°C. The resin was then washed three times with wash buffer (50 mM Tris, pH 7.4, 150 mM NaCl) and the bound proteins were eluted with SDS-PAGE sample buffer and analyzed by Western blotting.

## Glutathione-S-Transferase (GST) fusion protein interaction assay

Human 293 T cells in 6-well plates were co-transfected with 100 ng of GST or 1 µg of GST-Vpr expression plasmids and 500 ng of HA-tagged CCDC137 expression plasmids. The total amount of DNA was held constant by supplementing the transfection with empty expression vector. Two days later, cells were treated with MG 132 for 4 hr and then lysed in Lysis buffer (50 mM Tris, pH 7.4, 150 mM NaCl, 5 mM EDTA, 5% glycerol, 1% Triton X-100, and 1x complete protease inhibitor [Roche]). Cleared lysates were then incubated with glutathione sepharose (GE healthcare) for 4 hr at 4°C and, after wash with buffer (50 mM Tris, pH 7.4, 150 mM NaCl, 5 mM EDTA, 0.1% Triton X-100), bound proteins were eluted in SDS-PAGE sample buffer and subjected to Western blot analysis.

## BirA-fusion protein interaction assay

Human 293 T cells in 6-well plates were transfected with a V5-tagged CCDC137 expression plasmid and BirA (R118G) or BirA (R118G)-Vpr expression plasmid. Cells were treated with 50 µM biotin at 24 hr after transfection, and 10 µM MG132 at 40 hr post transfection, and harvested at 44 hr post

transfection. Cells were then lysed and cleared lysates were incubated with Dynabeads (MyOne Steptavadin C1) above. After thorough wash, biotinylated proteins were eluted from the beads with SDS-PAGE sample buffer supplemented with 200 µM biotin and subjected to Western blot analysis.

## Western blot analysis

Cell lysates and immunoprecipitates were separated on NuPage Novex 4–12% Bis-Tris Mini Gels (Invitrogen), and NuPAGE MES SDS running buffer (Invitrogen, NP0002) was used when Vpr was detected. Proteins were blotted onto nitrocellulose membranes. Thereafter, the blots were probed with primary antibodies and followed by secondary antibodies conjugated to IRDye 800CW or IRDye 680. Fluorescent signals were detected and quantitated using an Odyssey scanner (LI-COR Biosciences).

## Cell cycle analysis

To determine effects of Vpr on cell cycle, V1 based viral stocks were used to inoculate $2.5 \times 10^5$ U2OS cells in 6-well plates at an MOI of 0.5 to 1. At 48 hr post-infection, cells were trypsinized, fixed with paraformaldehyde (PFA) in phosphate-buffered saline (PBS), washed with PBS, and fixed again in 70% ethanol. After an additional wash with PBS, the cells were resuspended in FxCyclePI/RNase Staining Solution (Invitrogen) and incubated at 30℃ for 30 min. Flow cytometric analysis was performed using Attune NxT Acoustic Focusing Cytometer (ThermoFisher Scientific).

Alternatively, cells were transduced with pLKO.1-TRC-derived vectors encoding shRNA targeting CCDC137. After 48 hr, cells were selected in 1 µg ml$^{-1}$ puromycin or 5 µg ml$^{-1}$ blasticidin prior to propidium iodide staining, 40 hr later as described above.

In some experiments, U2OS cells expressing mKO2-hGeminin (1–110 aa) or mClover-hGeminin (1–110 aa) were transduced with LKO-derived lentiviral vectors encoding shRNAs targeting CCDC137. After 48 hr, cells were selected in 1 µg ml$^{-1}$ puromycin prior to FACS analysis, 48 hr later.

For the experiment in *Figure 6*, U2OS cells expressing doxycycline-inducible CCDC137 were generated by transduction with a LKO-derived lentiviral vector (*Busnadiego et al., 2014*) followed by selection in 1 µg ml$^{-1}$ puromycin. Cells were plated at the density of $2.5 \times 10^5$ in 6-well plates in the presence of doxycycline and the next day were infected with V1/HA-Vpr at an MOI of 2 or 0.5. At 12 hr post-infection, doxycycline was replenished and at 48 hr post-infection, cells were harvested for Western blot analysis (for cells infected at high MOI) or cell cycle analysis (for cells infected at low MOI).

## Fixed cell microscopy

U2OS cells expressing V5-tagged CCDC137 were seeded on 3.5 cm, glass-bottomed dishes coated with poly-L-Lysine (MatTek). At 48 hr after infection with V1/δ-Vpr or V1/HA-Vpr, cells were then fixed with 4% paraformaldehyde, permeabilized with 0.1% Triton X-100 and incubated with mouse anti-V5 monoclonal antibody (Invitrogen, R960-25) and rabbit anti-Fibrillarin polyclonal antibody (abcam, ab5821) followed by goat anti-mouse IgG Alexa fluor-594 conjugate and goat anti-rabbit IgG Alexa fluor-488 conjugate (Invitrogen). Images were captured using an DeltaVision OMX SR imaging system (GE Healthcare). Endogenous CCDC137 in 293 T cells or in macrophages was stained with rabbit anti-CCDC137 polyclonal antibody (abcam, ab185368) using the same procedure.

For detection of γ-H2AX foci, U2OS cells were transduced with lentiviruses encoding CCDC137-targeting shRNAs and, after selection with puromycin, cells were seeded on 3.5 cm, glass-bottomed dishes coated with poly-L-Lysine (MatTek). Nuclear foci were visualized by immunostaining with rabbit anti-γ-H2AX (abcam, ab11174) followed by a goat anti-rabbit IgG Alexa Fluor-594 conjugate (Invitrogen). A Z-series of images were acquired using an DeltaVision OMX SR imaging system (GE Healthcare).

## Fluorescence in situ hybridization (smFISH)

At 48 hr postinfection, macrophages in 8-well chamber slides were washed with PBS (Ambion), fixed with 4% formaldehyde (Thermo Fisher) in PBS and permeabilized with 70% ethanol. Then, the cells were washed with Stellaris RNA FISH wash buffer A (Biosearch Technologies) and GFP RNA

detected by incubation using 0.125 µM Cy5-labeled probes in Stellaris RNA FISH hybridization buffer (Biosearch Technologies) for 18 hr at 37℃. The cells were then washed twice in Stellaris RNA FISH wash buffer A (Biosearch Technologies) with the presence of Hoechst stain during the second wash. Then the cells were washed briefly with Stellaris RNA FISH wash buffer B (Biosearch Technologies), rinsed three times with PBS and subject to imaging by deconvolution microscopy (DeltaVision OMX SR imaging system). All images were generated by maximum intensity projection using the Z project function in ImageJ 1.52b. Corrected total cell fluorescence (CTCF) was calculated as the following: CTCF = Integrated Density - (Area of selected cell x Mean fluorescence of background readings). The 28 probes against GFP (cggtgaacagctcctcgc, gaccaggatgggcaccac, gtttacgtcgccgtccag, acacgctgaacttgtggc, gccggtggtgcagatgaa, ggtggtcacgagggtggg, actgcacgccgtaggtca, tcggggtagcggctgaag, agtcgtgctgcttcatgt, ggcatggcggacttgaag, ctcctggacgtagccttc, gccgtcgtccttgaagaa, tcggcgcgggtcttgtag, tgtcgccctcgaacttca, ctcgatgcggttcaccag, tgaagtcgatgcccttca, caggatgttgccgtcctc, cgttgtggctgttgtagt, gcttgtcggccatgatat, gtcctcgatgttgtggcg, gtagtggtcggcgagctg, cgatggggg tgttctgct, ttgtcgggcagcagcacg, gactgggtgctcaggtag, ttggggtctttgctcagg, catgtgatcgcgcttctc, ggtcacgaactccagcag, cttgtacagctcgtccat) were designed using the Stellaris Probe Designer, version 2.0 (Biosearch Technologies).

## Measurement of mRNA levels using qPCR

Macrophages ($6 \times 10^5$) were infected with V1/sh carrying shRNA targeting luciferase or CCDC137 (II), and 48 hr later RNA was isolated using the NucleoSpin RNA Kit (Macherey-Nagel). RNA levels were determined with Power SYBR Green RNA-to-CT 1-Step Kit using a StepOne Plus Real-Time PCR system (Applied Biosystems). For relative quantification, samples were normalized to actin. The sequences of primers are GAGCGCACCATCTTCTTCAA (GFP forward), TCCTTGAAGTCGATGCCCTT (GFP reverse), GAGCTAGAACGATTCGCAGTTA (Gag forward), CTGTCTGAAGGGATGGTTGTAG (Gag reverse) CATGTACGTTGCTATCCAGGC (actin forward) and CTCCTTAATGTCACGCACGAT (actin reverse). Relative GFP and Gag expression was calculated as the value of $2^{-[\Delta Ct (GFP)- \Delta Ct (actin)]}$.

## Live cell microscopy

To monitor cell cycle and HIV-1 (V1) gene expression infection in living cells, U2OS cells expressing mClover-hGeminin (1–110 aa) or primary macrophages were transferred into glass-bottom dishes and time-lapse microscopy was performed using a VivaView FL incubator microscope (Olympus). In some experiments, cells were transduced with lentiviruses containing shRNA targeting CCDC137, 36 hr prior to imaging. In some experiments, cells were infected with V1/δ-Vpr or V1/HA-Vpr expressing mCherry or GFP 12 to 24 hr prior to imaging. Images were captured every 30 min using GFP, mRFP and DIC filter sets for up to 72 hr. Preparation of movies was done using MetaMorph software (Molecular Devices) as previously described (*Holmes et al., 2015*). Images had a depth of 12 bits, that is, an intensity range of 0–4095.

## Replicates and statistics

All data is plotted raw, that is individual values for each individual quantitative determination is plotted. The exception to this is CCDC137/Vpr western blot data in *Figure 1B*, in which the mean of two independent experiments is plotted, with error bars representing the range of the duplicate raw values. Statistical comparisons between groups in *Figures 6C*, *8H* and *9E,F,G*. were done using Graphpad Prism software, and p-values were calculated using a Welch's t-test or a ratio t-test.

# Acknowledgements

We thank Proteomics Resource Center, Rockefeller University for mass spectrometry analysis. We thank Agata Smogorzewska, Theodora Hatziioannou and Trinity Zang for reagents and other members of the Bieniasz and Hatziioannou laboratories for helpful discussions.

## Additional information

### Funding

| Funder | Grant reference number | Author |
|---|---|---|
| National Institute of Allergy and Infectious Diseases | AIR3764003 | Paul D Bieniasz |
| Howard Hughes Medical Institute | | Paul D Bieniasz |

The funders had no role in study design, data collection and interpretation, or the decision to submit the work for publication.

### Author contributions

Fengwen Zhang, Conceptualization, Formal analysis, Investigation, Writing - original draft, Writing - review and editing; Paul D Bieniasz, Conceptualization, Supervision, Funding acquisition, Writing - original draft, Project administration, Writing - review and editing

### Author ORCIDs

Paul D Bieniasz (iD) https://orcid.org/0000-0002-2368-3719

### Decision letter and Author response
Decision letter https://doi.org/10.7554/eLife.55806.sa1
Author response https://doi.org/10.7554/eLife.55806.sa2

## Additional files

### Supplementary files
• Supplementary file 1. List of proximal/interacting proteins identified using Mass Spectrometry analysis of biotinylated proteins in BirA (R118G)-fused Vpr-expressing cells. The table includes UniProtKB accession number, description, mean peak area, scores (the sum of the highest ions score for each distinct peptide), percent coverage (calculated by dividing the number of amino acids in all found peptides by the total number of amino acids in the entire protein sequence), the number of distinct peptides used for identification in a protein, and peptide spectrum matches (PSM, the total number of identified peptide sequences for the protein). The table contains results from duplicate samples.

• Supplementary file 2. List of oligonucleotides and their sequences and cDNA and accession numbers employed in the molecular construction of expression plasmids used in this study.

• Transparent reporting form

### Data availability
All data generated or analysed during this study are included in the manuscript and supporting files.

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
