## [Decision Letter]

**Acceptance summary:**

The Vpr protein of HIV-1 remains the most enigmatic of the virus' accessory proteins. The main phenotypes attributed to Vpr are the ability to delay or arrest dividing cells in the G2/M phase of the cell cycle and to enhance viral replication in macrophages (which are non-dividing cells) and to a lesser extent in T cells. Vpr is recognized as a substrate adapter for a cullin-based ubiquitin ligase, but the substrate associated with G2-arrest has not yet been clearly identified. Zhang and Bieniasz now identify CCDC137 as the cellular protein whose Vpr-induced proteasomal degradation is responsible for G2/M arrest, which is a very nice step forward. They further show that mutants of CCDC137 that cannot bind Vpr inhibit Vpr-induced cell-cycle arrest, further establishing a causal relationship.

**Decision letter after peer review:**

Thank you for submitting your article "HIV-1 Vpr induces cell cycle arrest and enhances viral gene expression by depleting CCDC137" for consideration by *eLife*. Your article has been reviewed by three peer reviewers, and the evaluation has been overseen by Wes Sundquist as Reviewing Editor and Päivi Ojala as the Senior Editor. The following individuals involved in review of your submission have agreed to reveal their identity: Vicente Planelles (Reviewer #1); Michael Emerman (Reviewer #3).

Summary:

The Vpr protein of HIV-1 remains the most enigmatic of the virus' accessory proteins. The main phenotypes attributed to Vpr are the ability to delay or arrest dividing cells in the G2/M phase of the cell cycle and to enhance viral replication in macrophages (which are non-dividing cells) and to a lesser extent in T cells. Vpr is recognized as a substrate adapter for a cullin-based ubiquitin ligase, and several cellular targets of Vpr have been reported previously, but their association with Vpr function has generally been uncertain, although Vpr-induced activation of the SLX4-complex has been connected with G2-arrest, and Vpr induced degradation of HTLF has been associated with stimulation of viral growth rate in primary T cells.

Zhang and Bieniasz now identify CCDC137 as the cellular protein whose Vpr-induced degradation is responsible for G2/M arrest. Specifically, the authors used a variation of the BioID system to identify proximal proteins and find Ki67 as the top hit, although Ki67 is not degraded. They then hypothesize that a Ki67-interacing protein could be the direct target of Vpr. More than a dozen candidate Ki67-interacting proteins were tested, and CCDC137 was the only one that reproducibly showed Vpr-induced degradation. CCDC137 degradation could be abolished with a proteasome inhibitor (MG132) or by mutating residue Q65 in Vpr, which is required for interaction with DCAF1, a substrate receptor of Cullin 4. They also showed that shRNAs against DCAF1 abrogate degradation of CCDC137. The bulk of the evidence presented is consistent what we know about Vpr, including that Vpr stimulates viral LTR transcription modestly (3-5 fold) and that shRNA-mediated degradation of CCDC137 phenocopies Vpr activity in both inducing G2 arrest. Mutants of CCDC137 that cannot bind Vpr inhibited Vpr-induced cell-cycle arrest, further establishing a causal relationship. The effects are more pronounced in macrophages, in agreement with an early report from the Landau lab. Based on the fact that differentiated macrophages are not cycling and do not express ATR and other relevant cell cycle signaling molecules nor exhibit the hallmarks of Vpr-induced DNA damage, the authors logically speculate that cell cycle and genotoxic effects may be epiphenomena.

Overall, the main findings are well supported by the data, the core results are convincing, and this work represents a significant advance.

Essential revisions:

Although additional experiments are unnecessary, the authors should comment on the following issues:

1) The authors do not discuss their results fully in the context of published data about other Vpr targets, which should be done more completely in the Discussion, and in a more nuanced form. For example, in the Introduction, the statement that none of the other a potential Vpr targets has an effect on the G2 arrest is an over-statement. The Greenwood et al. paper (2020, Cell Rep.) showed that knockdown of several of the proteins that they identified as direct binders of Vpr including SMN1, CDCA2, and ZNF267 increased cells in G2. Were any of these among the candidates tested and found not to be reproducible?

2) A related point is that in first section of the Results: CCDC137 was already identified in the Greenwood et al. paper as a direct binder of both HIV-1 and HIV-2 Vpr. This paper was published in April 2019 and was online in bioRxiv since July 2018. This does not detract from the fine work in the present paper that validates this hit, but the authors should acknowledge up front that their screen found the same protein as another extensive one, and now they follow-up on one that was in common. Were there any other hits that were in common between the two screens?

3) Was the Vpr R80A mutation tested for interaction with and degradation of CCDC137? This mutant Vpr does not cause cell cycle arrest, and it acts as a dominant negative in a manner that is rescued by the mutation Q65R, which blocks binding to DCAF1. In other words, Vpr-80A should not bind the key cellular protein involved in cell cycle arrest. If this was not tested, please explain in the text any potential caveats to the present analysis.

4) Subsection “Endogenous CCDC137 depletion in HIV-1 infected cells”: Did SIVmac Vpr cause G2 arrest in these cells? Did HIV-2 Vpr cause a reduced G2 arrest? It's important to link the phenotype with the effects of each Vpr on CCDC137 levels.

5) Figure 6: The experiment showing that a Vpr-resistant version of CCDC137 attenuates the Vpr-mediated G2 arrest is nice. However, for this data to be convincing, it needs quantitation and some statistics (especially since the WT CCDC137 ectopic expression seems to have an effect relative to the "none").

6) The use of primary cells in the presented studies is commendable, but wild type virus is not used. The HIV-1 gene expression assays all use a minimal HIV construct that is not replication competent and is introduced after packaging and pseudotyping much as a lentiviral vector. Please address this potential shortcoming in the text.

7) Figure 5—video 1: It is very difficult to appreciate the reported effects, even if the main text tells you what to look for. Please comment.

---

## [Author Response]

Essential revisionsAlthough additional experiments are unnecessary, the authors should comment on the following issues:1) The authors do not discuss their results fully in the context of published data about other Vpr targets, which should be done more completely in the Discussion, and in a more nuanced form. For example, in the Introduction, the statement that none of the other a potential Vpr targets has an effect on the G2 arrest is an over-statement. The Greenwood et al. paper (2020, Cell Rep.) showed that knockdown of several of the proteins that they identified as direct binders of Vpr including SMN1, CDCA2, and ZNF267 increased cells in G2. Were any of these among the candidates tested and found not to be reproducible?

We concede that we did not sufficiently credit the Greenwood et al. paper, which was cited, but not discussed in detail. This has been addressed in the revised manuscript, in the Introduction and in the Discussion.

To address the specific question, SMN1, CDCA2, and ZNF267 whose depletion was reported have modest effects on cell cycle by Greenwood et al., we cotransfected fixed amount of plasmids expressing V5-tagged ZNF267, SMN or CCDC137 with various amounts of a Vpr expression plasmid (0, 25, 50 ng) in 293T cells (see Author response image 1). Based on the findings of Greenwood et al., SMN was the most interesting candidate as its depletion was reported to cause about 19-30% cells to become arrested in G2/M. However, as shown in the attached western blot, neither the canonical SMN isoform (294aa) nor a shorter isoform (282aa) was depleted by Vpr, under conditions where CCDC137 (289aa) was profoundly depleted. The expression level of ZNF267 in transfected cells was below the WB detection limit and therefore it is hard to reach a conclusion as to whether Vpr controls its stability. However the effects of ZNF267 depletion reported in the Greenwood et al. paper are extremely modest. We did not test CDCA2 (Repo-Man) as it is enriched in NK cells and Treg cells, but not in macrophages and therefore it is unlikely to be depleted by Vpr to enhance viral gene expression in macrophages. Notably, Trinkle-Mulcahy L et al., 2006, showed that depletion of Repo-Man (CDCA2) using RNA interference induces apoptosis with little G2/M accumulation (see Figure 5 of that paper). Moreover, the G2M accumulation reported by Greenwood et al. following ZNF267 depletion was also very modest.

As reported by many authors and by Greenwood et al., and others, many proteins can be depleted by Vpr, reflecting the fact that Vpr hijacks CRL4 ubiquitin ligase complex that governs the stability of hundreds of proteins. However, CCDC137 is the only protein thus far demonstrated to be depleted by extremely low levels of Vpr. The depletion of a number of proteins might contribute to cell cycle arrest to varying degrees, particularly under conditions of Vpr overexpression. However, we emphasize that the potency with which Vpr induces CCDC137 depletion and the profound effects of CCDC137 depletion on cell cycle and HIV-1 gene expression, particularly in macrophages, make this Vpr target unique.

2) A related point is that in first section of the Results: CCDC137 was already identified in the Greenwood et al. paper as a direct binder of both HIV-1 and HIV-2 Vpr. This paper was published in April 2019 and was online in bioRxiv since July 2018. This does not detract from the fine work in the present paper that validates this hit, but the authors should acknowledge up front that their screen found the same protein as another extensive one, and now they follow-up on one that was in common. Were there any other hits that were in common between the two screens?

We concede that we did not sufficiently credit the Greenwood et al. paper, which was cited, but not discussed in detail. This has been addressed in the Discussion of revised manuscript. The Greenwood et al. paper did not identify CCDC137 as a Vpr binding protein, rather it reported that the levels of many proteins (one of which was CCDC137) were decreased in HIV-1 or HIV-2 infected cells. No follow up experiments on CCDC137 were reported by Greenwood et al. and it could not be discerned whether the reduced CCDC137 levels was a cause or consequence of G2/M arrest.

Proteins that were found in both in our BirA screen and in the Greenwood et al. paper (2019) include GNL2 (high score), CCNT1 (cyclin T1, high score), UTP14A (low score), DNTTIP2(low score), and PINX1(low score). We tested the high scoring hits (GNL2 and CCNT1) but these proteins were not validated as bona fide Vpr targets in our follow up experiments. UTP14A knockdown has been reported to induce G1 arrest, no information of which we are aware exisits for DNTTIP2.

3) Was the Vpr R80A mutation tested for interaction with and degradation of CCDC137? This mutant Vpr does not cause cell cycle arrest, and it acts as a dominant negative in a manner that is rescued by the mutation Q65R, which blocks binding to DCAF1. In other words, Vpr-80A should not bind the key cellular protein involved in cell cycle arrest. If this was not tested, please explain in the text any potential caveats to the present analysis.

Preliminarily, we find that Vpr R80A retained some ability to deplete CCDC137 in transient overexpression essays in 293T cells, but not in infection assays in macrophages. Consistent with that finding, Vpr R80A failed to enhance viral gene expression in macrophages. An extensive analysis of Vpr mutants is underway (but cannot be completed at present) and will be the subject of future studies.

We cannot formally rule out the possibility that additional proteins whose level is reduced under high level Vpr expression might contribute to the G2/M arrest phenotype in transfected cells. We have modified the Discussion to concede this caveat. We note though, that our findings strongly suggest that the advantage conferred by Vpr is via the enhancement of viral gene expression rather than cell cycle arrest in G2/M.

4) Subsection “Endogenous CCDC137 depletion in HIV-1 infected cells”: Did SIVmac Vpr cause G2 arrest in these cells? Did HIV-2 Vpr cause a reduced G2 arrest? It's important to link the phenotype with the effects of each Vpr on CCDC137 levels.

The reviewers make a fair point. SIVmac Vpr has a minimal impact on cell cycle in infected human U2OS cells. HIV-2 Vpr caused G2 arrest to some degree. This data is shown in Figure 4—figure supplement 2 of the revised manuscript and is consistent with previous observations. Importantly, the effects of SIVmac and HIV-2 Vpr proteins on CCDC137 levels do indeed correlate with their abilities to induce cell cycle in human cells.

5) Figure 6: The experiment showing that a Vpr-resistant version of CCDC137 attenuates the Vpr-mediated G2 arrest is nice. However, for this data to be convincing, it needs quantitation and some statistics (especially since the WT CCDC137 ectopic expression seems to have an effect relative to the "none").

Conceded. Quantitation and statistical evaluation of the effects of WT and Vpr-resistant mutant CCDC137 overexpression on Vpr-induced cell cycle arrest are shown in Figure 6C of the revised manuscript.

6) The use of primary cells in the presented studies is commendable, but wild type virus is not used. The HIV-1 gene expression assays all use a minimal HIV construct that is not replication competent and is introduced after packaging and pseudotyping much as a lentiviral vector. Please address this potential shortcoming in the text.

We have modified the Discussion to address this caveat. However, we would point out that some experiments (Figure 8—figure supplement 2) were done with a near full-length proviral constructs (carrying GFP in place of Nef) that was (we concede) introduced into macrophages by infection using VSV-G pseudotyping.

7) Figure 5—video 1. It is very difficult to appreciate the reported effects, even if the main text tells you what to look for. Please comment.

We are puzzled by this comment. To us it is very clear that a higher proportion of the cells in the lower panels (transduced with CCDC137-depleting shRNA vector) have a higher number of cells with visible mClover geminin, indicating that they are in G2 (a frame from the video is in Author response image 2)

**Author response image 2. respfig2:**